# Membrane Charge Effects on Solute Transport in Nanofiltration: Experiments and Molecular Dynamics Simulations

**DOI:** 10.3390/membranes15060184

**Published:** 2025-06-18

**Authors:** Suwei Liu, Zihao Foo, John H. Lienhard, Sinan Keten, Richard M. Lueptow

**Affiliations:** 1Department of Mechanical Engineering, Northwestern University, Evanston, IL 60208, USA; suweiliu2017@u.northwestern.edu (S.L.); s-keten@northwestern.edu (S.K.); 2Department of Mechanical Engineering, Massachusetts Institute of Technology, Cambridge, MA 02139, USA; zihaofoo@alum.mit.edu (Z.F.); lienhard@mit.edu (J.H.L.); 3Center for Computational Science and Engineering, Massachusetts Institute of Technology, Cambridge, MA 02139, USA; 4Department of Civil and Environmental Engineering, Northwestern University, Evanston, IL 60208, USA; 5Department of Chemical and Biological Engineering, Northwestern University, Evanston, IL 60208, USA; 6The Northwestern Institute on Complex Systems (NICO), Northwestern University, Evanston, IL 60208, USA

**Keywords:** molecular dynamics, solute transport, water filtration, nanofiltration

## Abstract

Polyamide membranes, such as nanofiltration (NF) membranes, are widely used for water purification. However, the mechanisms of solute transport and solute rejection due to solute charge interactions with the membrane remain unclear at the molecular level. Here, we use molecular dynamics simulations to examine the transport of single-solute feeds through charged nanofiltration membranes with different membrane charge concentrations of ${\mathrm{COO}}^{-}$ and NH+2 resulting from the deprotonation or protonation of polymeric end groups according to the pH level that the membrane experiences. The results show that ${\mathrm{Na}}^{+}$ and ${\mathrm{Cl}}^{-}$ solute ions are better rejected when the membrane has a higher concentration of negatively charged groups, corresponding to a higher pH, whereas ${\mathrm{CaCl}}_{2}$ is well rejected at all pH levels studied. These results are consistent with those of experiments performed at the same pH conditions as the simulation setup. Moreover, solute transport behavior depends on the membrane functional group distribution. When ${\mathrm{COO}}^{-}$ functional groups are concentrated at membrane feed surface, ion permeation into the membrane is reduced. Counter-ions tend to associate with charged functional groups while co-ions seem to pass by the charged groups more easily. In addition, steric effects play a role when ions of opposite charge cluster in pores of the membrane. This study reveals solute transport and rejection mechanisms related to membrane charge and provides insights into how membranes might be designed to achieve specific desired solute rejection.

## 1. Introduction

Polyamide membranes for both reverse osmosis (RO) and nanofiltration (NF) have emerged as a crucial technology for addressing the growing issue of global water scarcity because of their efficient water treatment and desalination capabilities. As the demand for fresh water rises, driven by population growth, industrial activities, and climate change, NF membranes provide an attractive solution due to their unique ability to selectively remove divalent ions and large molecules with a low energy cost [1,2,3,4]. This selectivity is particularly beneficial for applications where the removal of divalent ions like calcium and magnesium is required, such as in brackish water desalination, wastewater treatment, or lithium recovery [5,6]. Compared to other membrane-based desalination technologies such as reverse osmosis (RO), which is necessary for monovalent ion removal [7,8], NF membranes offer advantages in terms of energy efficiency [6,9], as they operate at lower pressures and, thus, consume less energy while achieving effective ion rejection [10,11]. In many cases, membrane charge plays a critical role in determining the selectivity of NF membranes for different ion species through mechanisms such as electrostatic interactions and Donnan exclusion [12,13]. However, many fundamental aspects of ion–membrane charge interactions in polyamide membranes are not fully understood [14,15,16], particularly at the molecular level.

It is widely accepted that the pH of the feed solution can alter the ionization state of membrane functional groups, leading to changes in the membrane’s charged group density and the spatial distribution of charges [17,18]. Understanding the ion–membrane interactions is crucial for optimizing NF membrane design in terms of membrane heterogeneity and charge group modification [19,20]. Traditional experimental approaches, such as zeta potential measurements and ion rejection tests, provide valuable macroscopic insights [21,22,23,24] but lack the capability of capturing the complex interactions that govern solute ion transport through a membrane with charged groups [25,26,27,28] at the nanoscale, both in terms of time scales measured in nanoseconds and length scales measured in Ångstroms. Moreover, experimental studies present significant challenges in precisely controlling and characterizing the distribution of charged groups within NF membranes [29,30]. For instance, it is difficult to directly measure the changes in charge density as a function of membrane thickness through the active layer, which is only about 20 nm thick [31], or to visualize how different charged groups within the membrane impact ion transport.

Here, we attempt to overcome these experimental limitations by using molecular dynamics (MD) simulations to provide atomistic-level insights. MD simulations are a powerful tool to address this knowledge gap because it is possible to investigate the molecular interactions between solutes and the membrane under controlled conditions with resolution at the nanoscale. MD simulations reveal detailed information at the atomic scale, such as density/porosity measurement across membrane thickness at the Ångstrom level, the trajectories of individual solute ions, and the interaction energies between ions and the membrane charged groups. Hence, MD simulations can shed light on mechanisms that cannot be easily captured through experiments or theoretical approaches at the continuum level [32,33,34]. Previous MD simulations of RO membranes of trimesoyl chloride (TMC) polymerized with m-phenylene diamine (MPD) and NF membranes of TMC polymerized with piperazine (PIP) have proven valuable in unraveling the intricate mechanisms governing water and solute transport through these membranes at the molecular scale [35,36,37,38,39].

In this study we consider both monovalent (NaCl) and divalent (${\mathrm{CaCl}}_{2}$) feeds using single-solute feed solutions. The comparison between ${\mathrm{Na}}^{+}$ and ${\mathrm{Ca}}^{2+}$ ions allows us to investigate how various membrane charged groups influence the transport of solute ions at different pH levels, which is reflected in the local membrane charge distributions. This transport is particularly relevant for practical applications, where the ability to selectively reject divalent ions over monovalent ones is crucial for membrane-based water treatment processes such as ion–ion aqueous separations in water. Additionally, understanding how pH-induced changes in membrane charge affect this selectivity could inform strategies for tuning membrane properties to achieve desired separation outcomes.

To investigate membrane–ion charge interactions, we construct three virtual membrane models with different spatial distributions of charged groups in the membrane corresponding to three different pH levels, demonstrating how to implement both negative and positive charges to the membrane nanostructure in MD simulations. To further isolate the effects of charge concentration and distribution on ion transport, we construct five other membrane variations with different combinations of membrane charge distributions so that we can consider how these variations affect ion transport at pH = 7. Due to the limitations of computational resources, we apply a body force to solute ions in order to drive their movement through the membranes, while maintaining negligible transmembrane pressure. To provide a comparison to our computational results, we also conduct experiments with similar global settings to that of the MD simulations including feed solution concentration, temperature, and solute type. The experiments conducted here provide general guidance for interpreting the nanoscale computational results but they do not serve as direct comparisons because of the immense differences in time and length scales.

It is important to note that concepts related to membrane charge like Donnan exclusion and dielectric exclusion are continuum level effects based on the integrated effect of an immense number of individual charge interactions between ions in the feed solution and charged groups in the membrane [40]. In the MD simulations we describe here, we consider these individual molecular level electrostatic interactions at length and time scales that are many orders of magnitude smaller than these continuum level viewpoints. Likewise, buildup of feed ions in a concentration polarization layer at the membrane surface is measured at the scale of tens or hundreds of microns over seconds or minutes [41], whereas our MD simulations are at the scale of nanometers and nanoseconds. Thus, MD simulations reflect solute concentrations deep within concentration polarization layer within a few nanometers of the membrane surface. Furthermore, while concentration polarization may affect the overall rejection, comparative results for the nanoscale transport of ions within the membrane, which is the focus here, are unaltered by macroscale concentration polarization. In spite of the difficulty in connecting these vastly different length and time scales, we consider both simulations and experiments in this paper to advance the overall understanding of solute transport mechanisms in charged polyamide membranes, focusing on the critical role of membrane charge on solute transport. More specifically, this study serves as guidance for effective strategies for the spatial distribution of membrane charge to control ion transport and rejection that can be implemented to fabricate innovative nanofiltration membranes using a membrane-by-design approach. Additionally, the understanding of ion–surface interactions developed in this paper may have application to other types of membranes [25,42] as well as situations such as bio-inspired solid-state nanopores, biosensors, and energy-related interfaces. Recent advances in electrochemical and optical techniques, including single-molecule measurements, further underscore the importance of fully understanding ion-specific behavior at charged interfaces across a wide range of applications [43,44,45,46].

## 2. Methods

### 2.1. Membrane Model Preparation

When polyamide membranes are in operation, the feed pH and $pK_{a}$ determine the ionization state of amine groups (*R*-NH) and carboxyl groups (*R*-COOH) within the membrane. Hence, we use MD simulations to examine the effect of specific charge concentration on solute transport in a NF membrane at different pH levels. The first step is to build a neutral membrane model for subsequent membrane charge modifications. To start, an approximately stoichiometric mixture of 332 PIP (piperazine) and 228 TMC (trimesoyl chloride) monomers is randomly packed in a 5.2 × 5.2 × 5 nm^3^ simulation reaction box within which the monomers have the freedom to move due to Brownian motion. We use the “fix bond/react” command [47] in LAMMPS [48] to simulate polymerization using the canonical ensemble with a timestep of 1 fs. More specifically, whenever a H atom from PIP is within 3.5 Å of a Cl atom from TMC, these two atoms are deleted and an amide bond is formed between the two monomers. This virtual polymerization continues for 40 ps until the reaction sites reach a self-limiting state. Then, the bonding distance criteria is increased to 5.0 Å for the next 160 ps to speed up the polymerization reactions. Next, unreacted Cl atoms are replaced by hydroxyl groups, forming COOH end groups. The entire virtual membrane structure undergoes equilibration for 20 ns. A simplified illustration of the final model of the PIP-TMC dimer is shown on the left in Figure 1. Further details of the polymerization process and how the model membrane properties are consistent with those of actual NF membranes formed through interfacial polymerization can be found in our previous publication [49].

The main focus of this work is to investigate how differences in the membrane charge state affect solute transport, but it is quite difficult and computationally expensive to model the dynamic process of amine group protonation and carboxyl group deprotonation. Instead, we construct membranes with pre-defined ionization states for the solute transport simulations. In our previous work [49], we constructed negatively charged membranes by removing hydrogen atoms from selected carboxyl groups (*R*–COOH) to form *R*–${\mathrm{COO}}^{-}$. The number of deprotonated carboxyl groups depends on how far the feed pH is above the isoelectric point (IEP), as discussed in later sections. Here, we also construct positively charged membranes, where the number of amine groups (*R*–NH) that are protonated to *R*–NH+2 depends on how far the feed pH is below the IEP. Protonated and deprotonated dimers are shown on the right in Figure 1. The general AMBER force field (GAFF) [50] is used to describe the forcefields for NH+2, and AM1-BCC in AmberTools18 [51,52] is utilized to determine the topology and partial charges of ionized amine groups. Other simulation parameters are similar to those in our previous work [49].

To study the membrane charge concentration at different pH values, we utilize the relationship between the concentration of functional groups and the feed pH quantified by Ritt et al. [18] within the NF270 active layer. In such thin-film composite membranes, the active layer is a crosslinked polyamide thin film, typically formed via interfacial polymerization of PIP and TMC monomers. This active layer is the main focus of this study rather than the membrane support layers, which do not play a role in the separation process. Ritt et al. measured membrane charge concentrations for NH+2 and ${\mathrm{COO}}^{-}$ at 6 different pH levels and fit curves to their data for pH = 2 to 12, as reproduced in Figure 2. The NH+2 areal density decreases sharply when the feed pH increases above 8 and reaches zero for pH > 10, as indicated by the left vertical axis in Figure 2a. On the other hand, the ${\mathrm{COO}}^{-}$ areal density (left vertical axis in Figure 2b) starts at zero for pH < 2 and increases as feed pH increases. We select pH = 2, 7, and 10 as the main focus for this study, because these pH values represent three distinct ionization states of the membrane that correspond directly to measurements by Ritt et al. [18]. More specifically, at pH = 2, the positively charged membrane only has positive charged groups, at pH = 7, there is a combination of positively and negatively charged groups, and at pH = 10, only negatively charged groups are present. The membrane model at pH = 7 serves as an improved model compared to our previous one [49] in that protonated amine groups are included as well as deprotonated carboxyl groups. Note that we use the areal functional group densities corresponding to the experimental measurements by Ritt et al. [18], which correspond to data points in Figure 2, noting that they deviate slightly from the curve fits that Ritt et al. computed.

To convert the areal functional group densities (sites ${\mathrm{nm}}^{-2}$) to the number of functional groups within the approximately 100 nm^3^ volume of our membrane model, it is necessary to account for the membrane thickness, surface area, and charge distribution profile. First, the densest region of our membrane model spans about 3 nm in thickness, but an actual NF membrane active layer is about 7 times thicker (about 20 nm [31]). To convert the areal density found by Ritt et al. to the number of charged functional groups in the simulated membrane, we assume that the sites they measured were distributed through the thickness of a 20 nm thick membrane. For instance, for NH+2 at low pH, the areal density is 4.3 sites ${\mathrm{nm}}^{-2}$ in Figure 2a, which corresponds to a volume concentration of 0.2 ($\frac{4.3}{20}$) sites ${\mathrm{nm}}^{-3}$. For the densest portion of the simulated membrane, which has a volume of about 67 nm^3^ this corresponds to 14 sites for charged NH+2 groups (0.2 sites ${\mathrm{nm}}^{-3}$ × 67 nm^3^) in the simulated membrane, giving a net charge of +14*e* for the simulated membrane. This number of NH+2 groups is depicted on the right vertical axis in Figure 2a. For instance, at pH = 2, there are 14 NH+2 groups and no ${\mathrm{COO}}^{-}$ groups. In this case, we use a computer algorithm to randomly select 14 amine groups across the thickness of the membrane in *z*-direction and protonate them as shown in Figure 3a, where the qualitative distribution of NH+2 groups through the thickness of the membrane is shown by the green curve just above the membrane model.

At pH = 7, the areal NH+2 density is larger than the areal ${\mathrm{COO}}^{-}$ density, which would seemingly correspond to a membrane with a positive net charge. This contradicts the negative zeta potential measurement at pH = 7 that we measured for a commercial NF270 polyamide membrane (described in Section 2.3 and Section 3.1). This could be a consequence of the challenges related to measuring the number and distribution of charged groups through the thickness of real membranes using silver-binding/ICP-MS method in [18], where the charged group areal density is hard to measure with Ångstrom-level resolution across the thickness of NF270 active layer, which spans only about 20 nm. This also reflects the challenges in connecting Ångstrom-resolution simulations to macroscale membrane characterization.

To provide a negative net charge for the membrane at pH = 7, consistent with the results zeta potential measurements described in Section 3.1, while simultaneously matching areal densities of both membrane charged groups, we hypothesize that the areal density measurements reported by Ritt et al. are accessing different portions of NF270 across the active layer thickness direction for NH+2 and ${\mathrm{COO}}^{-}$. More specifically, we speculate that the NH+2 areal density reflects NH+2 groups across the entire thickness of the active layer, whereas the ${\mathrm{COO}}^{-}$ areal density reflects ${\mathrm{COO}}^{-}$ groups concentrated in the surface region of the active layer because of an enhanced negative surface charge resulting from the post-treatment procedures performed during NF270 fabrication [53]. At this point, we need to mention a different experimental study by Coronell et al. [17] that we relied on in our previous study of flux in charged membranes [49]. Based on the Coronell et al. study, we expect a net membrane charge at pH = 7 for the membrane model to be −14*e*, which is consistent with the negative zeta potential measured for a NF270 polyamide membrane at pH = 7 in Section 3.1.

To reconcile the 14 NH+2 charged groups at pH = 7 based on the Ritt et al. study (right vertical axis in Figure 2b) with the net membrane charge of −14*e* at pH = 7 based on the Coronell et al. study, we implement $28\;{\mathrm{COO}}^{-}$ groups in the simulated membrane at pH = 7. To take the membrane charge heterogeneity into account, we randomly deprotonate 28 carboxyl groups within a 1.5 nm thick layer of the membrane at the feed surface, while protonating 14 amine groups throughout the entire membrane model thickness, as shown in Figure 3b. In other words, the membrane model at pH = 7 can be viewed as a representation of the charge distribution in a physical membrane, since we attempt to represent the positive charge density near the feed surface (left side) in the 20 nm thick active layer of a real membrane within the much thinner membrane model that we can simulate. The distributions of carboxylate groups (yellow curve) and amine groups (green curve) through the thickness of the membrane are shown just above the membrane model in Figure 3b. Note that the local density of NH+2 is higher toward the permeate surface (right side), which matches the charge distribution of polyamide NF membranes fabricated through interfacial polymerization [54,55].

Continuing this approach that combines the findings of Ritt et al. [18] with those of Coronell et al. [17], at pH = 10 there should be $30\;{\mathrm{COO}}^{-}$ groups concentrated near the feed surface of the membrane model and no NH+2 groups at all. However, because the membrane model is so thin compared to an actual membrane, we randomly deprotonate 30 COOH groups spanning the entire membrane thickness, as shown in Figure 3c. This distribution of negative charges serves as a contrast to pH = 2 where positive charges are located throughout the membrane rather than concentrated in a local region as in the pH = 7 scenario. This membrane model can also be thought of as representing the feed surface “slice” of a physical membrane at pH = 10, omitting the remainder of the active layer of the membrane, which has neutral amine groups and a small number of ${\mathrm{COO}}^{-}$ groups.

Although it is necessary to make several assumptions, these membrane models exhibit a range of membrane charge distributions that allows the study of electrostatic charge interactions under diverse conditions. This makes possible the detailed exploration of how various membrane charge conditions influence the structural and functional properties of NF membranes, particularly the interactions of membrane charge with ions in the feed solutions, which is the focus of this study.

To further investigate charge interactions between ionic solutes and membrane charge, we also consider five other versions of membrane models for pH = 7. While the net charge of these membrane models is always −14*e*, there are different ways to allocate the charged functional groups across the membrane thickness, as shown in Figure 3d–h. The “flip” model is the mirrored image of the model shown in Figure 3b; the “bilayer” model separates the positive and negative functional charged groups into two layers where the carboxylate groups are all concentrated at the feed surface and the ionized amine groups are all located near the opposite surface. The slightly different distributions of charged groups in Figure 3b,d come about during membrane equilibration with slightly different effects of charge-balancing counter-ions, as described shortly.

We also revisit our previously constructed pH = 7 model [49], in which only $14\;{\mathrm{COO}}^{-}$ groups are present, distributed in various ways across the membrane model thickness, to result in a net −14*e* net charge, as shown in Figure 3f–h. We allocate these $14\;{\mathrm{COO}}^{-}$ to either the surface of the membrane or at the center of the membrane. Note that the −*z* direction is toward the feed reservoir, and the +*z* direction is toward the permeate reservoir, which corresponds to the model naming convention indicated in Figure 3f–h.

For all the models shown in Figure 3, the *x*- and *y*-directions are the spanwise directions with periodic boundaries, while the *z*-direction is non-periodic through the thickness of the membrane with the feed on the left and the permeate on the right. The membrane models are illustrated in *yz*-plane to clearly show the charged groups, whereas the analyses in later sections display the *xz*-plane. There is no structural difference between these two planes. After setting the protonation or deprotonation states of the membrane at a specific pH level, the membrane undergoes hydration and equilibration similar to in our previous study [49]. To offset the membrane charge during equilibration, counter-ions (not shown in Figure 3) are added in equal amounts to the feed and permeate reservoirs to ensure correct electrostatics calculations. More specifically, for a positively charged membrane (+14*e*) at pH = 2, $14\;{\mathrm{Cl}}^{-}$ ions are used to render a zero net charge in the simulation domain. For negatively charged membranes (−14*e* or −30*e*), the counter-ion species depends on the solute feed. For example, when a pH = 7 membrane (−14*e*) is used to examine its interaction with a NaCl feed, $14\;{\mathrm{Na}}^{+}$ ions are added to balance the membrane charge, whereas if simulating with a ${\mathrm{CaCl}}_{2}$ feed, $7\;{\mathrm{Ca}}^{2+}$ ions are added as counter-ions instead. During equilibration, the transmembrane pressure is zero and counter-ions are allowed to move freely into the membrane. After equilibration, the counter-ions are typically in vicinity of membrane charged groups due to electrostatic attraction. Further details of membrane preparation can be found in our previous publication [49].

The average density of all membrane models used in this study is 0.80 ± 0.03 g ${\mathrm{cm}}^{-3}$ across the densest regions. We also characterize pore sizes using PoreBlazer v4.0 [56] through a geometric method that determines the total free volume and the pore size distribution given a spherical “probe” of a particular size, yielding a mean pore diameter of 5.6 ± 0.3 Å, although there are both network and aggregate pores consistent with previous simulations [49] and experimental characterization [57]. The density and mean pore diameter values match with the “loose” membrane models and experimental results reported in our previous publication [49]. A detailed characterization of all membrane models used in this study is available in Appendix A, which tabulates net charge, charged group composition, density, and porosity for each membrane model.

### 2.2. Solute Transport Simulations

To explore the intricate interplay between charged solute behavior and membrane charge, both NaCl and ${\mathrm{CaCl}}_{2}$ are used as the solutes. This allows for the examination of both monovalent and divalent cations and their interactions with the charged membranes. The solute transport system that we simulate is shown in Figure 4. The system comprises one of the membrane models shown in Figure 3 with reservoirs on either side of the membrane having the same size (5.2 × 5.2 × $4.6\;{\mathrm{nm}}^{3}$) and matching the dimensions of the membrane in the *x*- and *y*-directions with periodic boundary conditions. A single-layer graphene sheet is positioned on the outside of each reservoir on which external pressure of 0.1 MPa is applied, resulting in zero transmembrane pressure. In addition to the water molecules already present in the membrane during hydration and equilibration, the feed and permeate reservoirs are filled with 3918 water molecules packed at 1 g ${\mathrm{cm}}^{-3}$. The water reservoirs and graphene sheets are equilibrated for 10 ns and the entire system undergoes a minimization process before production runs. Since our focus is on comparing monovalent versus divalent cation transport, we use the same concentrations of ${\mathrm{Na}}^{+}$ and ${\mathrm{Ca}}^{2+}$ ions. In the feed reservoir, there are either $10\;{\mathrm{Na}}^{+}$ and $10\;{\mathrm{Cl}}^{-}$ or $10\;{\mathrm{Ca}}^{2+}$ and $20\;{\mathrm{Cl}}^{-}$, with the original counter-ions used to equilibrate the membrane remaining in the simulation to balance the charge. Given the size of the feed reservoir, this particular number of solute ions is equivalent to 0.133 M NaCl or ${\mathrm{CaCl}}_{2}$ solution. In the permeate reservoir, only water molecules are present initially. The counter-ions added during the equilibration process are initially in vicinity of membrane charges; however, they can move freely during simulations. After a typical 100 ns simulation, most counter-ions remain within 5 Å of the membrane charged groups.

Our preliminary simulations, along with another study [58], have demonstrated that with transmembrane pressure alone, solute feed ions cannot penetrate the membrane within a reasonable amount of simulation time (75 ns at a transmembrane pressure of 600 MPa). Hence, drawing upon a technique previously utilized to study ion transport [39,59,60], we use a body force generated by an imposed electric field in the positive *z*-direction to bias solute ions to transit from the feed reservoir on the left to the permeate reservoir on the right within a time range consistent with computational constraints. No pressure gradient is applied across the membrane. Reminiscent of electrodialysis, this computational technique merely serves to speed up the movement of solutes into and through the membrane during the simulation. To ensure an appropriate body force, we considered electric field voltages from 0.1 V to 1.0 V in increments of 0.1 V. An electric field of 0.5 V is adequate to drive all feed ions into the membrane without clustering at the membrane surface and, while in the membrane, allowing them to interact with charged end groups and find pathways through the membrane (negligible ion displacement is observed for ≤0.3 V) without disrupting the membrane structure or rupturing the membrane chains (membrane deformation occurred for ≥0.6 V). This 0.5 V electric field results in a body force of 0.072 kcal ${\mathrm{mol}}^{-1}{Å}^{-1}$ applied to monovalent ions (${\mathrm{Na}}^{+}$ and ${\mathrm{Cl}}^{-}$) and 0.144 kcal ${\mathrm{mol}}^{-1}{Å}^{-1}$ for divalent ions (${\mathrm{Ca}}^{2+}$). To allow the membrane to move within a small range of motion without being displaced due to force-biased solute ions, roughly 5% of the atoms in the *yz*-plane at the periodic boundaries of the membrane are fixed in space. To obtain statistically meaningful data, each membrane model is simulated for 100 ns (6 days of wall-clock time for two simulations running in parallel on a high-performance workstation equipped with 40 CPUs and an NVIDIA^®^ Quadro^®^ GV100 GPU (NVIDIA Corporation, Santa Clara, CA, USA). Note that here, we focus on investigating the solute–membrane interactions rather than water transport, which we considered previously [49]. Because the transmembrane pressure is zero and, subsequently, no water flows through the membrane, the reservoir sizes remain constant during the simulation.

For all the MD simulations performed in this study, we use the NAMD simulation package [61] along with general AMBER forcefields (GAFF) [50]. The interactions for NaCl and ${\mathrm{CaCl}}_{2}$ are based on Lennard-Jones and Coulombic terms to capture van der Waals and electrostatic contributions [62]. The SHAKE algorithm [63] with a cutoff of the non-bonded potential of 9 Å is used to constrain the bond between each hydrogen and its mother atom. The particle mesh Ewald (PME) method [64] is used to compute full electrostatics with a grid spacing of 1 Å. The time step is set at 1 fs with output saved every 2 ps. The water model used here is TIP4P [65], which results in reasonable water transport performance and ion dynamics in NF membrane simulations [66]. The global temperature is set to 300 K, where the Langevin damping coefficient constant is $2\;{\mathrm{ps}}^{-1}$. Each case has three runs for a total of 48 independent solute transport simulations.

### 2.3. Experiment Design

Since membrane models are polymerized using PIP and TMC, which are the major components of the commercial FilmTec NF270 membrane (DuPont Water, Edina, MN, USA) active layers, we use NF270 as benchmark to compare with the computational results. To provide context for the virtual membrane model and to demonstrate the relationship between membrane charged group densities and feed pH, we characterize the NF membrane charge under various feed pH values via tangential streaming potential experiments. The NF membrane is acquired from a commercial spiral wound module (DuPont NF270-2540). A 0.1 M KCl streaming solution is used in a 100 μm gap cell (Anton Paar GmbH, Graz, Austria), where the solution pH ranges from 2 to 10. The zeta potentials are calculated from the streaming potential measurements with the classical Smoluchowski equation [67,68].

In addition, coupon-scale NF experiments are conducted with a plate-and-frame cross-flow membrane module, as described in our prior publication [24]. An 8 cm × 3 cm channel with a thickness of 1 mm is utilized, and a cross-flow with a mean velocity of 8.5 cm $s^{-1}$, corresponding to a feed flow rate of about 2.5 mL $s^{-1}$, is sustained with a positive displacement pump (Hydra-Cell F20, Minneapolis, MN, USA). The NF membrane and the feed and permeate spacers are acquired from a commercial spiral wound module (DuPont NF270-2540). The pressure pulsations introduced during circulation are mitigated with pulsation dampeners (Hydra-Cell 4CI SST). The feed stream pressure, conductivity, temperature, solution pH, and permeate mass are monitored in real-time with LabView and are recorded at 1 Hz frequency.

Anhydrous ACS-grade NaCl and ${\mathrm{CaCl}}_{2}$ salts are procured from MilliporeSigma and dissolved in Type 1 ultrapure water (18.2 MΩ cm) to prepare the NaCl and ${\mathrm{CaCl}}_{2}$ feed solutions at a cation concentration of 0.130 M, and the feed solution pH is adjusted using 1 M HCl and 1 M NaOH. The NF membranes are compacted with ultrapure water at 40 bar for 6 h before experiments with inorganic salt solutions are conducted. In each inorganic salt experiment, the membrane coupon is equilibrated with the feed solution for 15 min at a transmembrane pressure of 10 or 15 bar before feed and permeate samples are collected. In these experiments, the transmembrane water fluxes were approximately three orders of magnitude lower than the crossflow fluxes. After equilibration, three samples of the feed and permeate solutions are collected at the same time from the feed inlet and permeate outlet tubes, respectively, and chilled in centrifuge tubes. The retentate stream is recirculated back into the feed reservoir, which has a negligible effect on the feed concentration due to the volume of the feed reservoir. The inorganic composition of the samples is determined with inductively coupled plasma optical emission spectroscopy (Agilent ICP-OES 5100, Santa Clara, CA, USA), using a five-point calibration curve based on standards from MilliporeSigma (Trace-Cert). The ion permeation is based on the concentration measurements with ICP-OES as(1)$$P_{i}=\frac{C_{i,p}}{C_{i,f}}\times 100\%,$$
where $P_{i}$ denotes the permeation of ion *i* (-), and $C_{i,f}$ (M) and $C_{i,p}$ (M) denote the concentration of ion *i* in the feed and permeate streams, respectively.

## 3. Results and Discussion

### 3.1. Experimental Results

We begin with the experimental results to provide context for the more extensive simulation results. The zeta potential dependence on pH for a NF270 membrane is shown in Figure 5. Based on a second-order polynomial interpolation, the isoelectric point (IEP) of the NF270 membrane is at a solution pH of 3.3. Although zeta potential measurements have limitations and assumptions with the measurements only reflecting the net surface charge, which is influenced by a number of factors, the zeta potential provides a useful measure of the *relative* surface charge behavior. Below the IEP, the amine groups in the active layer are generally protonated in the acidic environment and have a positive charge while the carboxyl groups are protonated and neutral. Hence, the NF270 membrane has a net positive charge density as a result of the positively charged NH+2 groups [18]. Conversely, the carboxyl groups are more likely to be deprotonated with increasing pH levels, particularly above the pKa of carboxylic acids, which is around 4–5, consistent with increasingly negative zeta potentials for increasing pH in Figure 5. At pH = 7, both NH+2 and ${\mathrm{COO}}^{-}$ groups coexist, although there is a greater abundance of ${\mathrm{COO}}^{-}$ groups, since TMC (trimesoyl chloride) contains three acid chloride groups, while PIP (piperazine) contributes only two amines. This 3:2 stoichiometric imbalance inherently leads to a net negative charge, consistent with the zeta potential at pH = 7. Finally, at higher pH, ${\mathrm{COO}}^{-}$ groups dominate, leading to an even lower zeta potential. Hence, the character of the NF membrane zeta potential is consistent with the virtual membrane models described in Section 2.1, which have a net positive charge at pH = 2 and an increasingly negative charge at pH values of 7 and 10.

The ion permeation results from the experiments with NaCl and ${\mathrm{CaCl}}_{2}$ feed solutions are presented in Figure 6a,b, respectively, and tabulated in Appendix A. The rate of cation and anion permeation is constrained by electro-neutrality requirements in the permeate solution [18,24]. The permeation of ${\mathrm{Na}}^{+}$ and ${\mathrm{Cl}}^{-}$ ions in the NaCl experiments increases from 50.1 % at pH = 2 to a local maximum of 79.1 % at pH = 7, before decreasing to 70.1 % at pH = 10. The same non-monotonic trend in permeation has been reported for symmetric salts (cations and anions of the same charge valency, e.g., NaCl, ${\mathrm{CaSO}}_{4}$) [69,70]. As a result of the inherent transport coupling between cations and anions, the permeation rate of symmetric salts is limited by the kinetics of the co-ion transport [70,71]. Here, a co-ion is defined as the ion with the same charge as that of the active layer of the membrane. For instance, at solution pH above the IEP (here, at pH = 3.3), the charge density of the negatively charged ${\mathrm{COO}}^{-}$ groups increases with increasing pH [18]. Hence, it is usually assumed that the anion permeation rate decreases with feed solutions at higher pH, as observed here when pH increases from 7 to 10, while cation permeation decreases to maintain charge balance. Similarly, an equivalent relationship between the cation transport and the solution pH is observed, where the cation permeation rate decreases at low solution pH from the higher density of the positively charged NH+2 groups. At the same time, anion permeation decreases to maintain charge balance. The ion permeation rate is maximized at the isoelectric point when Coulombic repulsion and Donnan exclusion are at their weakest with a net neutral membrane, giving rise to the distinct inverted V-shape relationship between ion permeation and solution pH [69].

With asymmetric salts, which have cations and anions of unequal charge valencies (${\mathrm{CaCl}}_{2}$), the kinetics of ion permeation is governed by the ion with the higher charge valency [70]. Our experiments with ${\mathrm{CaCl}}_{2}$ feed solutions agree with this observation. The ion permeation of ${\mathrm{Ca}}^{2+}$ and ${\mathrm{Cl}}^{-}$ increases with the solution pH, in conjunction with the weakening of Donnan exclusion on ${\mathrm{Ca}}^{2+}$ ions at higher solution pH levels from the reduced density of ionized amine groups [18]. As observed from our experiments in Figure 6b, the permeation of ${\mathrm{Ca}}^{2+}$ increases monotonically from 7.4 % at pH = 2 to 57.8 % at pH = 10, while the zeta potential of the active layer declines monotonically from +15 mV to −51 mV over the same pH interval. In essence, our experimental measurements with symmetric and asymmetric salts align with previous results [69]. We further note that our intent here is to consider pH values that reflect relevant protonation regimes of the membrane, from fully protonated NH+2 groups at pH = 2 to fully deprotonated ${\mathrm{COO}}^{-}$ groups at pH = 10, rather than to explore the pH values corresponding to maximum or minimum salt rejection.

### 3.2. Solute Transport Simulations

#### 3.2.1. Monovalent Solute Transport

While experimental studies can measure the overall solute transport over the course of minutes-long experiments, all-atomistic MD simulations offer the opportunity for exceptionally detailed measurements of ion transport at nanosecond and nanometer scales. On the other hand, MD simulations can only sample a limited number of ion passage events, so direct comparison of permeance *$P_{i}$* between MD and experiments is difficult. The great advantage of MD simulations is that it is possible to consider the interactions of individual solute ions with charged groups in the membrane to provide a molecular-level picture of mechanisms pertaining to ion transport.

To elucidate the molecular level transport behavior of solute ions, we consider 0.133 M NaCl and 0.133 M ${\mathrm{CaCl}}_{2}$ solutions at three pH levels (pH = 2, 7 and 10) across three trials. The feed ion locations are depicted in the *xz*-plane at four time instances after starting in the reservoir on the feed side of the membrane, as shown in the first four columns in Figure 7 and Figure 8. In order to clearly indicate ions that have permeated completely through the membrane, we reset the *z*-locations for any ions with *z* > 40 Å to *z* = 40 Å. The gray shading in the first four columns indicates the local density of the membrane model, with darker gray corresponding to higher density. Ion and charged end group locations are projected onto the *xz*-plane, although they are at varying positions in the depthwise *y*-direction. Note that the positions of the charged end groups may shift slightly in position from one time instant to another. This is a consequence of the natural motion of the membrane nanostructure due to thermal vibrations and collisions with water molecules [37]. Also note that the vertical *x*-axis is stretched compared to the horizontal *z*-axis to more clearly show the distribution of ions and charged end groups. The last columns in Figure 7 and Figure 8 show the probability density function (PDF) of feed ion and membrane functional group *z*-locations at 100 ns based on a default Matlab kernel smoothing function (we used 50% of default bandwidth for ${\mathrm{Cl}}^{-}$ and 80% for the cations’ default bandwidths) [72,73,74,75].

First, we examine the transport for the 0.133 M NaCl solution ($10\;{\mathrm{Na}}^{+}$ and $10\;{\mathrm{Cl}}^{-}$) at pH = 2, as shown in Figure 7A. The results for all three trials are overlaid in the figure, corresponding to $30\;{\mathrm{Na}}^{+}$ and $30\;{\mathrm{Cl}}^{-}$ ions. Near the start of the simulation (*t* = 1 ns shown in Figure 7$A_{1}$), most ions are near the feed surface of the membrane, although ${\mathrm{Cl}}^{-}$ ions are closer to the membrane and one ${\mathrm{Cl}}^{-}$ (red circle) has already traveled past *z* = −20 Å to reside close to an NH+2 group (green star) due to electrostatic attraction. (In spite of a two-dimensional illustration here, we always confirm nearby locations, accounting for the *y*-direction as well). ${\mathrm{Na}}^{+}$ ions (black triangles) tend to be a little further from the membrane surface. By *t* = 10 ns (Figure 7$A_{2}$), ${\mathrm{Cl}}^{-}$ ions make further progress into the membrane, and two ${\mathrm{Cl}}^{-}$ ions are already past the center of the membrane (*z* = 0 Å). The majority of ${\mathrm{Cl}}^{-}$ ions inside membrane are in the vicinity of NH+2 groups (accounting for the *y*-direction as well). Most of the cations still reside on the surface of the membrane at this stage. However, some ${\mathrm{Na}}^{+}$ ions move into the membrane and are likely to be found near ${\mathrm{Cl}}^{-}$ ions either near membrane surface or inside the membrane (confirmed in the *y*-direction, although this is not immediately evident in the projection shown in Figure 7$A_{2}$). This is likely because NH+2 groups interact repulsively with ${\mathrm{Na}}^{+}$ ions, while ${\mathrm{Cl}}^{-}$ ions proceed quicker into the membrane due to their attraction with NH+2. As ${\mathrm{Cl}}^{-}$ ions move through the membrane, they are likely to chaperone ${\mathrm{Na}}^{+}$ along to maintain the local charge balance. Although this is not immediately evident in Figure 7A, the trajectory video of the entire transport process clearly suggests that some ${\mathrm{Cl}}^{-}$ ions carry ${\mathrm{Na}}^{+}$ ions along, following roughly the same path.

At *t* = 50 ns, one of the $30\;{\mathrm{Cl}}^{-}$ ions has reached the permeate reservoir, and $26\;{\mathrm{Cl}}^{-}$ ions are in the dense core region within the membrane, with three ${\mathrm{Cl}}^{-}$ ions remaining in the feed reservoir near the membrane surface. The ${\mathrm{Cl}}^{-}$ ions within the membrane are more likely to associate with NH+2 groups than ${\mathrm{Na}}^{+}$ ions, with a large number near *x* = 0 Å. In this region, ${\mathrm{Cl}}^{-}$ ions are likely to hop from one NH+2 group to another, based on careful observation of ion trajectory videos. In contrast, many ${\mathrm{Na}}^{+}$ ions are still in the feed reservoir at *t* = 50 ns, with only six of them making it past the membrane center typically in the vicinity of their nearby ${\mathrm{Cl}}^{-}$ ions. Although ${\mathrm{Na}}^{+}$ ions can be chaperoned by ${\mathrm{Cl}}^{-}$ ions as mentioned previously, we do not frequently observe such ${\mathrm{Na}}^{+}$ transport, most likely because the chaperoning effect is diminished by the strong repulsion between ${\mathrm{Na}}^{+}$ ions and NH+2, which is often referred to as Donnan repulsion [4,76,77,78]. In fact, two thirds of the ${\mathrm{Na}}^{+}$ ions have not even reached the dense region of the membrane yet.

At the end of the simulation (*t* = 100 ns), three ${\mathrm{Cl}}^{-}$ ions have reached the permeate and about $18\;{\mathrm{Cl}}^{-}$ ions have made it past the membrane center. Meanwhile, $12\;{\mathrm{Na}}^{+}$ ions have not even penetrated into the membrane surface, and only one ${\mathrm{Na}}^{+}$ ions has permeated through the entire membrane. It also appears that ion–ion interactions may hinder transport, particularly in the region of (*x*, *z*) = (0, 10), where ${\mathrm{Na}}^{+}$ and ${\mathrm{Cl}}^{-}$ ions appear to form clusters, thereby increasing steric repulsion of the clustered ions. More specifically, in this case, there are four ${\mathrm{Cl}}^{-}$ ions and four ${\mathrm{Na}}^{+}$ ions clustered together on average for each simulation trial within a region with a radius of 10 Å. At this point in the simulation (*t* = 100 ns), there are more ${\mathrm{Cl}}^{-}$ ions in the permeate reservoir than ${\mathrm{Na}}^{+}$ ions (three ${\mathrm{Cl}}^{-}$ and one ${\mathrm{Na}}^{+}$), corresponding to a net negative charge in the permeate.

The effect of charged NH+2 groups in the membrane on cations and anions at pH = 2 is evident in Figure 7$A_{5}$. The ${\mathrm{Na}}^{+}$ distribution at 100 ns is bi-modal. One peak of the ${\mathrm{Na}}^{+}$ distribution is located on the loose surface of the membrane (*z* = −25 Å) to the left of the maximum in the NH+2 distribution as the positively charged NH+2 groups act as a barrier for ${\mathrm{Na}}^{+}$ penetration beyond the membrane surface. However, it is likely that at times, ${\mathrm{Cl}}^{-}$ ions facilitate ${\mathrm{Na}}^{+}$ ions to overcome the NH+2 repulsion, resulting in the second peak in the ${\mathrm{Na}}^{+}$ distribution. The ${\mathrm{Na}}^{+}$ ions that have made it past the concentrated region of NH+2 groups overlap a dominant peak in the ${\mathrm{Cl}}^{-}$ distribution located near *z* = 10 Å, where ${\mathrm{Na}}^{+}$ and ${\mathrm{Cl}}^{-}$ ions seem to cluster. Given the ion positions shown in Figure 7$A_{1}$–$A_{4}$, it seems that at times, ${\mathrm{Cl}}^{-}$ transport takes place via NH+2 hopping due to attractive interactions. Furthermore, in addition to the dominant ${\mathrm{Cl}}^{-}$ peak near *z* = 10 Å, a second one overlapping with the densest region of NH+2 is likely a consequence of ${\mathrm{Cl}}^{-}$ ions associating with NH+2 groups. The far right peak indicates that some ${\mathrm{Cl}}^{-}$ ions have made it to the permeate reservoir.

As the pH increases to 7, there are ${\mathrm{COO}}^{-}$ groups distributed near the membrane feed surface, marked as yellow diamonds in Figure 7B. These negative charges near the membrane surface significantly change the ion–membrane and ion–ion interactions compared to pH = 2. By *t* = 10 ns, shown in Figure 7$B_{2}$, not only have three ${\mathrm{Cl}}^{-}$ ions made it past the middle of the membrane thickness, but one ${\mathrm{Cl}}^{-}$ ion has already permeated through the entire membrane. Conversely, ${\mathrm{Na}}^{+}$ ions engage with ${\mathrm{COO}}^{-}$ groups, as indicated by many ${\mathrm{Na}}^{+}$ ions being within 3 Å of a ${\mathrm{COO}}^{-}$ group. At this point, none of the ${\mathrm{Na}}^{+}$ ions bypass the ${\mathrm{COO}}^{-}$ layer due to the attractive interaction. As time advances to 50 ns, one third of the ${\mathrm{Cl}}^{-}$ ions are already in the permeate reservoir and another third of the ${\mathrm{Cl}}^{-}$ ions have penetrated past the middle of the membrane thickness. For ${\mathrm{Na}}^{+}$ ions, there is no significant transport into the membrane due to the ${\mathrm{Na}}^{+}$–${\mathrm{COO}}^{-}$ interaction. Although it may seem odd that ${\mathrm{Cl}}^{-}$ ions have penetrated through the membrane while the ${\mathrm{Na}}^{+}$ ions have not, overall electroneutrality has not been violated. It is simply that ${\mathrm{Cl}}^{-}$ ions are near the NH+2 charged group, and ${\mathrm{Na}}^{+}$ ions are near ${\mathrm{COO}}^{-}$ groups.

The difference between ${\mathrm{Na}}^{+}$ and ${\mathrm{Cl}}^{-}$ transport persists until the simulations end at *t* = 100 ns, where two thirds of the ${\mathrm{Cl}}^{-}$ ions have made it past *z* = 0 and none of the ${\mathrm{Na}}^{+}$ ions have penetrated beyond the ${\mathrm{COO}}^{-}$ layer. Although ${\mathrm{Na}}^{+}$–${\mathrm{COO}}^{-}$ interactions seem to have reached a quasi-steady state as significant advancement of ${\mathrm{Na}}^{+}$ is not observed in the *z*-direction, ${\mathrm{Cl}}^{-}$ ions tend to “hop” from one NH+2 to another in order to traverse through the membrane, while seven ${\mathrm{Cl}}^{-}$ ions remaining stuck near the membrane feed surface. It is likely that ${\mathrm{Na}}^{+}$ ions on the surface reduce the chance of ${\mathrm{Cl}}^{-}$ accessing NH+2 due to ${\mathrm{Na}}^{+}$–${\mathrm{Cl}}^{-}$ attractive interactions.

The results of charge interactions for pH = 7 are summarized in Figure 7$B_{5}$. The ${\mathrm{Na}}^{+}$ distribution has one peak near membrane feed surface, which mostly overlaps with the ${\mathrm{COO}}^{-}$ distribution, demonstrating how negative membrane charges hinder the transport of monovalent cations via attractive interactions. While ${\mathrm{COO}}^{-}$ groups repel solute anions, which might potentially hinder ${\mathrm{Cl}}^{-}$ transport in this case, many ${\mathrm{Cl}}^{-}$ ions permeate through the membrane, probably due to NH+2 groups deep in the membrane assisting ${\mathrm{Cl}}^{-}$ transport to overcome the repulsive barrier on the membrane feed surface. Hence, the ${\mathrm{Cl}}^{-}$ distribution is skewed so that its peak is located in the permeate reservoir.

At pH = 10, there are no NH+2 groups in the membrane. As *t* advances to 10 ns, shown in Figure 7$C_{2}$, ${\mathrm{Na}}^{+}$ ions penetrate into the membrane faster than ${\mathrm{Cl}}^{-}$ ions such that two ${\mathrm{Na}}^{+}$ ions are already near the center of membrane (*z* = 0 Å), and $23\;{\mathrm{Na}}^{+}$ ions have penetrated the membrane surface. In comparison, 17 of $30\;{\mathrm{Cl}}^{-}$ ions are still in the feed reservoir. For the ${\mathrm{Cl}}^{-}$ ions that have penetrated the membrane surface, they are most likely to be found in the vicinity of ${\mathrm{Na}}^{+}$ ions (accounting for the *y*-direction not shown here) due to electrostatic attraction. At *t* = 50 ns (Figure 7$C_{3}$), more ${\mathrm{Na}}^{+}$ ions advance further into the membrane but the *z*-locations for ${\mathrm{Cl}}^{-}$ ions do not advance significantly compared to *t* = 10 ns. At the end of the simulation (Figure 7$C_{4}$), one ${\mathrm{Na}}^{+}$ ion reaches the permeate reservoir and all other ${\mathrm{Na}}^{+}$ ions are within the membrane, usually in the vicinity of ${\mathrm{COO}}^{-}$ groups, accounting for all three dimensions. From simulation videos, it is evident that ${\mathrm{Na}}^{+}$ ions sometimes move along paths where ${\mathrm{COO}}^{-}$ groups are located as they traverse through the membrane in the *z*-direction. This suggests a hopping mechanism where ${\mathrm{Na}}^{+}$ ions jump from one ${\mathrm{COO}}^{-}$ to another due to electrostatic attraction. On the other hand, ${\mathrm{Cl}}^{-}$ ions inside the membrane are often found near ${\mathrm{Na}}^{+}$ ions and make little progress through the membrane, suggesting that ${\mathrm{Na}}^{+}$ facilitates ${\mathrm{Cl}}^{-}$ transport, opposite the scenario at pH = 2. Furthermore, the attractive charge of NH+2 end groups deep in the membrane drawing ${\mathrm{Cl}}^{-}$ ions through the membrane at pH = 2 and 7 is absent at pH = 10, thereby reducing ${\mathrm{Cl}}^{-}$ transport. These trends are reflected in the probability distributions illustrated in Figure 7$C_{5}$ where the ${\mathrm{Na}}^{+}$ distribution spans most of the membrane, but its peak is near the feed surface. On the other hand, ${\mathrm{Cl}}^{-}$ ions tend to be near the feed surface with the peak in the ${\mathrm{Cl}}^{-}$ distribution located in the feed reservoir.

It is clear that monovalent cations and anions interact differently with the membrane depending on the pH level. In an acidic feed (pH = 2), assisted by the positive membrane charge, ${\mathrm{Cl}}^{-}$ ions move faster than ${\mathrm{Na}}^{+}$ ions and some of the ${\mathrm{Cl}}^{-}$ ions reach the permeate reservoir within 100 ns. While ${\mathrm{Na}}^{+}$ transport is hindered by NH+2 groups through repulsion, ${\mathrm{Na}}^{+}$ ions can be chaperoned by ${\mathrm{Cl}}^{-}$ ions to traverse through the membrane. However, the ${\mathrm{Na}}^{+}$–${\mathrm{Cl}}^{-}$ clusters formed inside membrane eventually slow down ${\mathrm{Cl}}^{-}$ transport due to steric repulsion. As the pH increases to 7, ${\mathrm{Cl}}^{-}$ ions still have faster transport than ${\mathrm{Na}}^{+}$ ions. Since the ${\mathrm{COO}}^{-}$ layer on the membrane feed surface prevents ${\mathrm{Na}}^{+}$ ions from traversing further into the membrane, ${\mathrm{Cl}}^{-}$ ions that diffuse past the membrane surface are less likely to form clusters with ${\mathrm{Na}}^{+}$ ions. This leads to more ${\mathrm{Cl}}^{-}$ permeation than at pH = 2. However, in a basic solution at pH = 10, ${\mathrm{Cl}}^{-}$ transport is hindered by the negative membrane charge, but ${\mathrm{Na}}^{+}$ transport is facilitated by ${\mathrm{COO}}^{-}$ groups. This phenomenon is consistent with experimental results for ion transport in vicinity of co- and counter-ions [79].

A major challenge in comparing simulation results with experimental results is the duration of permeation. It is only possible to simulate the membrane for 100–200 ns, whereas the experiments are performed over times measured in minutes. In fact, the residence time for feed solution in the experiments is about 1 s, which is $10^{7}$ times longer than the entire simulation time of 100 ns. Thus, the experiments represent steady state permeation results for a membrane with ions dispersed through the entire membrane for a long time, whereas the simulations depict transient results over only 100 ns with feed ions starting in the feed solution. As noted in Section 2.2, another limitation related to the short duration of the simulations as well as the high energy barriers to ions is that a body force is necessary to push ions into the membrane within a reasonable simulation time rather than depending on pressure-driven flux of water and ions. Finally, as noted in Section 2.1, it is computationally difficult and expensive to model the dynamic process of amine group protonation and carboxyl group deprotonation. In spite of these differences between the simulations and the experiments, we comment on the relation between the simulation and experimental results.

Across the pH values that we consider, pH = 7 results in the most ${\mathrm{Cl}}^{-}$ permeation events within the course of the simulation, consistent with experimental observations in Figure 6a. The situation is less clear for ${\mathrm{Na}}^{+}$ at pH = 7 and for both ions at the other two pH levels, where membrane models at pH = 2 produce the second highest permeation, instead of pH = 10 as observed in experiments. We believe that this discrepancy may be related to how the membrane charge profile is modeled at the nanoscale, particularly the distribution of charged groups in the membrane. As discussed in Section 2.1, to qualitatively capture the charge profile within the membrane at different pH levels, we need to make some assumptions for pH = 7 and pH = 10 to best align the simulated NH+2 and ${\mathrm{COO}}^{-}$ distributions in the membrane with experimental findings while optimizing computational efficiency. However, because the distribution of ${\mathrm{COO}}^{-}$ and NH+2 groups through the membrane thickness is not clear from experiments, the membrane models used in this study may not exactly match the charge distributions of the membranes used in the experiments, which could cause the simulation permeation results to differ from the experimental results.

Despite the difficulty in directly comparing seconds-long experiments to nanoseconds-long simulations, the simulations here successfully demonstrate here how membrane charge and its distribution interact with the solute feed ions. Furthermore, we are able to consider other distributions of charged groups later in this paper to better understand how the distribution of charge in the membrane affects the results.

#### 3.2.2. Divalent Solute Transport

We also examine divalent cation transport by using 0.133 M ${\mathrm{CaCl}}_{2}$ feed solutions ($10\;{\mathrm{Ca}}^{2+}$ and $20\;{\mathrm{Cl}}^{-}$), with overlapping results for three trials shown in Figure 8. At pH = 2, ${\mathrm{Cl}}^{-}$ exhibits faster transport than ${\mathrm{Ca}}^{2+}$ starting from *t* = 1 ns (Figure 8$A_{1}$). More specifically, four of the $60\;{\mathrm{Cl}}^{-}$ ions quickly penetrate the low-density surface layer due to the NH+2–${\mathrm{Cl}}^{-}$ attraction, whereas all ${\mathrm{Ca}}^{2+}$ ions are still in the feed reservoir. By *t* = 10 ns, $53\;{\mathrm{Cl}}^{-}$ ions are inside the membrane and 13 of them have made it past the membrane center. Most are within 5 Å of NH+2 groups, accounting for all directions. Meanwhile, $25\;{\mathrm{Ca}}^{2+}$ ions are at the membrane feed surface, closely associated with ${\mathrm{Cl}}^{-}$ ions inside the membrane (also confirmed in *yz*-plane, not shown here).

At *t* = 50 ns, a cluster of ${\mathrm{Ca}}^{2+}$ and ${\mathrm{Cl}}^{-}$ ions forms in the region of *z* = 5 to 15 Å and *x* = −10 to 0 Å. It is likely that ${\mathrm{Cl}}^{-}$ ions first traverse to that region due to favorable ${\mathrm{Cl}}^{-}$–NH+2 interactions. Then those ${\mathrm{Cl}}^{-}$ ions further facilitate ${\mathrm{Ca}}^{2+}$ transport, which eventually leads to the formation of ${\mathrm{Ca}}^{2+}$–${\mathrm{Cl}}^{-}$ ion clusters. More evidence of the clustering is the near balance of charges for the ions—there are $32\;{\mathrm{Cl}}^{-}$ ions and $14\;{\mathrm{Ca}}^{2+}$ ions located in this small region. Furthermore, the local membrane density in this region is 0.2 g ${\mathrm{cm}}^{-3}$ (also indicated by its light gray color, most evident in Figure 7$A_{1}$ around (*z*, *x*) = (0, 0)), compared to an average membrane density of 0.8 g ${\mathrm{cm}}^{-3}$. This suggests the presence of a void that temporarily traps ${\mathrm{Ca}}^{2+}$ and ${\mathrm{Cl}}^{-}$ ions. By the end of the simulation at 100 ns, neither ion species has permeated through the membrane and nearly 60% of the solute feed ions cluster in the void, near (*x*, *z*) = (−5, 10). The peaks of ${\mathrm{Ca}}^{2+}$ and ${\mathrm{Cl}}^{-}$ distributions, as shown in Figure 8$A_{5}$, align with each other at this location, further demonstrating how many ions cluster at this location.

As the pH increases to 7, the ion transport differs from the case at pH = 2 starting at *t* = 10 ns (Figure 7$B_{2}$). First, ${\mathrm{Ca}}^{2+}$ ions spread out on the membrane surface, most likely due to ${\mathrm{Ca}}^{2+}$–${\mathrm{COO}}^{-}$ attraction. Additionally, ${\mathrm{Cl}}^{-}$ ions are closely associated with ${\mathrm{Ca}}^{2+}$ ions; only four ${\mathrm{Cl}}^{-}$ ions have made it past the membrane center. By the end of the simulation, shown in Figure 8$B_{4}$, only $11\;{\mathrm{Cl}}^{-}$ ions traverse past the membrane center, of which four ${\mathrm{Cl}}^{-}$ ions have permeated through the membrane. As shown in Figure 8$B_{5}$, the distributions for ${\mathrm{Ca}}^{2+}$ and ${\mathrm{COO}}^{-}$ overlap with nearly all of the ions centered near the membrane feed surface (near *z* = −20 Å). This further indicates that the ${\mathrm{COO}}^{-}$ groups on the membrane surface act as a charged barrier layer. For ${\mathrm{Cl}}^{-}$ ions, there are two peaks on either side of the ${\mathrm{COO}}^{-}$ peak with smaller ${\mathrm{Cl}}^{-}$ peaks beyond the ${\mathrm{COO}}^{-}$ peak.

At pH = 10, the *z*-locations for ${\mathrm{Ca}}^{2+}$ and ${\mathrm{Cl}}^{-}$ ions do not change significantly after *t* = 10 ns, as illustrated in Figure 8$C_{2}$–$C_{4}$. Most feed ions are located at *z* ≈ −20 Å and none of them traverse past *z* = 0 Å. Based on observing the trajectory videos, it seems that the ${\mathrm{COO}}^{-}$ groups first attract ${\mathrm{Ca}}^{2+}$ ions to the feed surface, to which ${\mathrm{Cl}}^{-}$ ions subsequently associate. Since there are no NH+2 groups present at pH =10, there appears to be a higher energy barrier for ${\mathrm{Cl}}^{-}$ ions to advance any further into the membrane, unlike the scenario at pH = 7. This is evident as the overlapping ${\mathrm{Ca}}^{2+}$ and ${\mathrm{Cl}}^{-}$ distributions at the membrane feed surface well to the left of the peak in the ${\mathrm{COO}}^{-}$ distribution, suggesting steric effects related to clusters of ${\mathrm{Ca}}^{2+}$ and ${\mathrm{Cl}}^{-}$ ions.

#### 3.2.3. Comparing Mono- and Di-Valent Solute Transport

Based on the results in Figure 7 and Figure 8, different transport behaviors are evident for monovalent cations (${\mathrm{Na}}^{+}$) and divalent cations (${\mathrm{Ca}}^{2+}$), which also affect the monovalent anions (${\mathrm{Cl}}^{-}$). In general, ${\mathrm{Na}}^{+}$ transits the membrane more easily than ${\mathrm{Ca}}^{2+}$. It appears that ${\mathrm{Na}}^{+}$ is less strongly associated with ${\mathrm{Cl}}^{-}$, evident as less overlap in the ${\mathrm{Na}}^{+}$ and ${\mathrm{Cl}}^{-}$ distributions, hence experiencing less the steric repulsion. To confirm this, we measure the ensemble average ion pair maximum lifetime for both feed solutions, which is based on consecutive time steps that a pair of ions are within 5 Å [80] of each other. For ${\mathrm{Na}}^{+}$–${\mathrm{Cl}}^{-}$, the ion pair lifetimes are 4.3 ns, 1.9 ns, and 9.0 ns at pH = 2, 7, and 10, respectively. On the other hand, the ion pair lifetimes for ${\mathrm{Ca}}^{2+}$–${\mathrm{Cl}}^{-}$ are an order of magnitude longer at 73.6 ns, 99.0 ns, and 98.9 ns for at pH = 2, 7, and 10. It seems likely that the strong association between ${\mathrm{Ca}}^{2+}$ and ${\mathrm{Cl}}^{-}$ ions leads to the formation of ${\mathrm{Ca}}^{2+}$–${\mathrm{Cl}}^{-}$ ion clusters, the sizes of which are greater than the average pore size of the membrane (about 5.6 Å). Hence, steric repulsion may play an important role of slowing solute transport for ${\mathrm{CaCl}}_{2}$ feed solutions. Of course, the number of ${\mathrm{Cl}}^{-}$ ions that are available is smaller for NaCl than for ${\mathrm{CaCl}}_{2}$, which may also play a role. Regardless, ${\mathrm{Na}}^{+}$ ions are still attracted to ${\mathrm{Cl}}^{-}$ ions, resulting in ${\mathrm{Na}}^{+}$ being transported more easily than ${\mathrm{Ca}}^{2+}$ via ${\mathrm{Cl}}^{-}$ chaperoning through the membrane. Of course, during the nanosecond scale of our simulations, the chaperoning effect is in competition with membrane charge interactions. Cation transport is hindered when ${\mathrm{COO}}^{-}$ groups are present due to attractive electrostatic interactions. The hindrance effect is most pronounced when ${\mathrm{COO}}^{-}$ groups are concentrated on membrane surface at pH = 7. On the other hand, ${\mathrm{Cl}}^{-}$ ions seem to be able to traverse the membrane via hopping from one NH+2 to another when NH+2 groups are spread out within a membrane at pH = 2 or pH = 7, but ${\mathrm{Cl}}^{-}$ transport is hindered at pH = 10 due to the lack of NH+2 groups.

To provide a clearer description of the ion transport through the membrane, Figure 9 shows the ensemble-average feed ion *z*-locations at 1 ns intervals throughout the entire simulation duration across three trials at each pH value for the first 100 ns of the simulations and for an additional 100 ns for a single trial at selected pH values. The slight discontinuity at 100 ns for the extended simulation time cases is a consequence of considering only a single trial for t>100 ns rather than the average of three trials for t<100 ns. Although the counter-ions added during the equilibration process to balance charge are initially in vicinity of membrane charges, they can move freely during simulations. After 100 ns, most counter-ions still remain within 5 Å of the membrane charges. Since they only play the role of ensuring electrostatics balance, their movement is not investigated here.

As shown in Figure 9a for NaCl, the average positions for both ions start in the feed reservoir around *z* = −30 Å and reach quasi-steady state positions in the membrane within the first 50 ns at all pH levels. Not only does this indicate that the simulation duration used in this study is sufficient to observe ion movement, it suggests that a transient phase occurs in which ions associate with charged end groups in the membrane and with each other, followed by a steady state phase in which charge interactions and steric effects inhibit ion transport through the membrane. However, it is important to note that the feed ions do not stop moving after they reach the quasi-steady state plateau for t>50 ns. The ions remain in constant motion, but their net progress through the membrane is negligible.

The ${\mathrm{Cl}}^{-}$ and ${\mathrm{Na}}^{+}$ ions exhibit different behaviors at each pH level. For pH = 2 and 7, ${\mathrm{Cl}}^{-}$ ions advance into the membranes faster, as indicated by the steep slope for the ${\mathrm{Cl}}^{-}$*z*-locations, and much further on average than ${\mathrm{Na}}^{+}$ ions. On the other hand, the average penetration of ${\mathrm{Na}}^{+}$ ions into the membrane is more limited, although the distributions of the ${\mathrm{Na}}^{+}$ ions in the membrane at steady state are much broader at pH = 2 and 10 than at pH = 7, as is evident in Figure 7. The net result is that the average *z*-locations for ${\mathrm{Na}}^{+}$ and ${\mathrm{Cl}}^{-}$ ions at the end of the simulation differ substantially. This suggests that ${\mathrm{Cl}}^{-}$–NH+2 attraction plays a significant role of assisting ${\mathrm{Cl}}^{-}$ movement during the transport process for the first 30–40 ns. A very different situation occurs for pH = 10, where ${\mathrm{Na}}^{+}$ ions penetrate further and faster into the membrane than ${\mathrm{Cl}}^{-}$ ions. Here the ${\mathrm{COO}}^{-}$ groups are positioned deeper in the membrane. They attract the ${\mathrm{Na}}^{+}$ ions but repel the ${\mathrm{Cl}}^{-}$ ions. Unlike at the lower pH levels, there are no ${\mathrm{Cl}}^{-}$–NH+2 interactions to draw ${\mathrm{Cl}}^{-}$ ions into the membrane.

For ${\mathrm{CaCl}}_{2}$, the average positions for both ions start in the feed reservoir around *z* = −30 Å and reach quasi-steady state positions in the membrane within the first 50 ns at all pH levels, as shown in Figure 9b. However, the differences between anion and cation *z*-locations are quite small compared to those for NaCl. This further corroborates the results in Figure 8, where ${\mathrm{Cl}}^{-}$ ions tend to travel together with ${\mathrm{Ca}}^{2+}$ due to electrostatic attraction. ${\mathrm{Ca}}^{2+}$ and ${\mathrm{Cl}}^{-}$ ions have the greatest penetration into the membrane at pH = 2 compared to pH = 7 and 10. This further indicates that ${\mathrm{COO}}^{-}$ groups near the feed surface at pH = 7 and 10 act as an electrostatic barrier for ${\mathrm{Ca}}^{2+}$ ions, while ${\mathrm{Ca}}^{2+}$ ions can travel farther into the membrane at pH = 2, where there is no electrostatic barrier. Nevertheless, the formation of ${\mathrm{Ca}}^{2+}$–${\mathrm{Cl}}^{-}$ ion clusters eventually limits the movement of both ions to about halfway into the membrane for *t* > 50 ns in all cases, as indicated by the similar penetration of ${\mathrm{Ca}}^{2+}$ and ${\mathrm{Cl}}^{-}$ ions. It is notable that the ${\mathrm{Cl}}^{-}$–NH+2 interactions at pH = 2 seem to be similar for NaCl and ${\mathrm{CaCl}}_{2}$ based on the similar ${\mathrm{Cl}}^{-}$ penetration in the two cases in spite of the larger number of ${\mathrm{Cl}}^{-}$ ions in the ${\mathrm{CaCl}}_{2}$ case. However, the penetration of ${\mathrm{Na}}^{+}$ ions is much less than that for ${\mathrm{Ca}}^{2+}$ ions.

Overall, the simulation results indicate that ${\mathrm{CaCl}}_{2}$ feed solutions at all pH levels have significantly less permeance compared to NaCl feed solutions, consistent with experimental results in Figure 6. For membrane models simulated using NaCl feeds, a feed at pH = 7 leads to the greatest ${\mathrm{Cl}}^{-}$ permeation, which agrees with our experimental data. However, the simulation results for ${\mathrm{CaCl}}_{2}$ at different pH levels seem at odds with the permeability experiments in Figure 6b. For example, ${\mathrm{Ca}}^{2+}$ and ${\mathrm{Cl}}^{-}$ ions move much further into the membrane at pH = 2 than at pH = 7 or 10, where they tend to get stuck at the feed surface of the membrane. Thus, one might expect the ${\mathrm{Ca}}^{2+}$ and ${\mathrm{Cl}}^{-}$ permeability to be higher at pH = 2 than at pH = 7 or 10, but the opposite is the case. We speculate that the discrepancy is a consequence of the vast difference in timescales and membrane thicknesses between the experiments and the simulations.

#### 3.2.4. Solute Transport Dependence on Charge Distribution

To further understand the effects of ion–membrane charge interactions on monovalent ion transport, we consider five other variations of membrane charge distributions (see Figure 3d–f). All of these membrane models have a net charge of −14*e*, corresponding to pH = 7. The difference resides in membrane charge species and membrane charge locations. We use the same solute transport simulation setup and compare the transport results directly with “control” membrane models shown in Figure 7.

How ${\mathrm{COO}}^{-}$ groups hinder ${\mathrm{Na}}^{+}$ ion transport is shown in Figure 10, where we vary the locations of ${\mathrm{COO}}^{-}$ groups. For example, in the “flip” membrane model, the NH+2 and ${\mathrm{COO}}^{-}$ groups are flipped in the *z*-direction compared to the “control” case so that the high concentration of ${\mathrm{COO}}^{-}$ group is on the permeate side of the membrane rather than the feed side (Figure 10a,b). NH+2 groups are also flipped but somewhat more dispersed across the membrane. The ions that are constrained at the feed surface at *t* = 100 ns are reversed in the two cases. That is, ${\mathrm{COO}}^{-}$ groups on the feed surface in the control case prevent ${\mathrm{Na}}^{+}$ ions from moving further into the membrane, while NH+2 groups near the feed surface in the flip case prevent ${\mathrm{Cl}}^{-}$ ions from moving further into the membrane. Furthermore, in spite of the ${\mathrm{Na}}^{+}$–NH+2 repulsion in the flip case, ${\mathrm{Na}}^{+}$ ions spread across the membrane toward the permeate reservoir. This suggests that the NH+2 layer on the membrane feed surface has minimal impact on ${\mathrm{Na}}^{+}$ transport. Instead, for the “flip” case, the collective ${\mathrm{Na}}^{+}$–${\mathrm{COO}}^{-}$ electrostatic attraction promotes ${\mathrm{Na}}^{+}$ ions advancing into the membrane, as will be shown later. Likewise, the ${\mathrm{COO}}^{-}$ layer in the control case seems to have a stronger impact on ${\mathrm{Na}}^{+}$ ions than on ${\mathrm{Cl}}^{-}$ ions. This result indicates that the impact of attractive electrostatic charge is greater for impeding the transport of counter-ions than the impact of repulsive electrostatic charge on impeding the transport of co-ions.

The phenomenon of a feed surface layer of ${\mathrm{COO}}^{-}$ trapping ${\mathrm{Na}}^{+}$ ions also occurs for the “bilayer” (Figure 10c) and “feed” (Figure 10d) charge distributions. When ${\mathrm{COO}}^{-}$ groups are distributed elsewhere, such as in the membrane “center” or toward the “permeate” reservoir (Figure 10e,f, respectively), some ${\mathrm{Na}}^{+}$ ions permeate through the membrane, and ${\mathrm{Na}}^{+}$ ions are less likely to get stuck on the membrane surface. Interestingly, the bilayer case and the feed case yield very different results even though both have a high concentration of ${\mathrm{COO}}^{-}$ near the feed surface. In the bilayer case, the attraction between ${\mathrm{COO}}^{-}$ groups and ${\mathrm{Na}}^{+}$ ions seems to trap the ${\mathrm{Na}}^{+}$ ions near the feed surface, while ${\mathrm{Cl}}^{-}$ ions pass through the membrane, aided by the collective electrostatic attraction of the NH+2 groups on the permeate side of the membrane. In the feed case, ${\mathrm{Na}}^{+}$ ions are trapped near the ${\mathrm{COO}}^{-}$ groups, but ${\mathrm{Cl}}^{-}$ ions are also trapped in the same vicinity. Curiously, a small number of ${\mathrm{Na}}^{+}$ and ${\mathrm{Cl}}^{-}$ ions permeate through the entire membrane. It is difficult, however, to draw general conclusions because of the presence of NH+2 in some cases and the difference in the overall number of ${\mathrm{COO}}^{-}$ groups in the different cases (28 vs. $14\;{\mathrm{COO}}^{-}$ groups in the “bilayer” and “feed” membrane models, respectively). Moreover, when local density of ${\mathrm{COO}}^{-}$ groups is higher on the surface in the feed case, ${\mathrm{Na}}^{+}$–${\mathrm{Cl}}^{-}$ clusters appear to form on the membrane surface, which may induce steric repulsion to further block pathways for other ${\mathrm{Na}}^{+}$ ions to pass through. Lastly, NH+2 and ${\mathrm{COO}}^{-}$ groups have different impacts on ${\mathrm{Cl}}^{-}$ transport. When NH+2 is present, a peak in the ${\mathrm{Cl}}^{-}$ distribution is associated with where NH+2 is concentrated (Figure 10a–c). When NH+2 is not present (Figure 10d–f), ${\mathrm{Cl}}^{-}$ tends to associate with its counter-ion, ${\mathrm{Na}}^{+}$.

We have shown previously that a ${\mathrm{CaCl}}_{2}$ feed solution results in few events where ions permeate entirely through the membrane. Not surprisingly, the results are similar for the other five membrane variations in Figure 11b–f. That is, a majority of the ${\mathrm{Ca}}^{2+}$ and ${\mathrm{Cl}}^{-}$ ions get stuck near the feed surface of the membrane and remain there throughout the entire simulation, regardless of the way charges are distributed in the membrane. This indicates a steric mechanism that prevents ${\mathrm{Ca}}^{2+}$ ion transport. This also corroborates the idea that the strong association between ${\mathrm{Ca}}^{2+}$ and ${\mathrm{Cl}}^{-}$ ions leads to clusters, which may also lead to steric effects that reduce the transport of both ions.

Again, we consider the ensemble-average feed ion *z*-locations at 1 ns intervals across three trials at each pH value for the different charge configurations. As shown in Figure 12 for both NaCl and ${\mathrm{CaCl}}_{2}$, the ions reach a quasi-steady state before the end of the 100 ns simulation, but the results are influenced by the charge distribution in the membrane, particularly for NaCl. For example, the flip and control charge configurations have nearly opposite results because of opposite charges on the feed side of the membrane. For the control case, ${\mathrm{Cl}}^{-}$ ions progress deep into the membrane drawn by the higher concentration of NH+2 charged groups, while ${\mathrm{Na}}^{+}$ ions penetrate only as far as the ${\mathrm{COO}}^{-}$ groups near the feed surface. In contrast, for the flip case, ${\mathrm{Na}}^{+}$ ions are drawn deep into the membrane by the ${\mathrm{COO}}^{-}$ groups, while the transport of ${\mathrm{Cl}}^{-}$ ions is restricted by the NH+2 groups nearer the feed surface. Interestingly, the broader distribution of NH+2 groups in the control case draws ${\mathrm{Cl}}^{-}$ ions further into the membrane than the narrower distribution of NH+2 groups near the permeate side of the membrane in the bilayer case.

The position of the ${\mathrm{COO}}^{-}$ charged groups in the membrane affects the transport significantly in the feed, center, and permeate cases in Figure 12j–l. The feed case has the least transport for both ions. In the feed and center cases, ${\mathrm{Na}}^{+}$ ions are drawn deeper into the membrane on average than ${\mathrm{Cl}}^{-}$ ions by the ${\mathrm{COO}}^{-}$ charged groups, but both ions penetrate the same distance in the permeate case, perhaps due to steric effects, before reaching the relatively distant ${\mathrm{COO}}^{-}$ groups as the ions move left to right. The ions travel together in all of the cases for ${\mathrm{CaCl}}_{2}$ but do not penetrate very far into the membrane past the feed surface.

### 3.3. Solute–Membrane Dynamics and Interactions

#### 3.3.1. Ion–Membrane Interaction Energy

To isolate charge interactions between the ions and the charged groups in the membrane from steric effects, we consider the interaction energies between individual solute feed ions and the membrane as the ions traverse through the membrane. The non-bonding interactions, including the dominant electrostatic interactions as well as van der Waals contributions, are computed using the NAMDEnergy tool, evaluated every 0.01 ns throughout the course of the simulation (100 ns). The energetics here represent the interaction between each solute ion and the collective membrane charge integrated over all of the charged end groups in the membrane, even though each charged end group acts as a point charge with the electrostatic force varying with the inverse of the distance squared. The computations assume a vacuum, as if no water is present. Neither steric effects nor water solvation, which can contribute to steric effects, are taken into account. This methodology provides a simplified yet straightforward characterization of ion–membrane interaction energy at the molecular level, omitting the complexities related to the aqueous environment. As a result, the interaction energy differences are larger than the actual free energy differences that are in play.

The interaction energy of each feed ion with the membrane charge as the ion travels in the *z* direction over the entire 100 ns simulation is shown in Figure 13 for the membrane models in Figure 3a–c, corresponding to different values of the pH. The interaction energies for each ion can only be computed at the *z*-locations that it has sampled. Therefore, regions without data indicate that ions were unable to permeate to these locations. The interaction energies for all ions in the three trials for the system are overlaid in the figure. The distributions of NH+2 and ${\mathrm{COO}}^{-}$ groups through the membrane thickness are also shown for reference.

Consider first the interaction energy for ${\mathrm{Na}}^{+}$ and ${\mathrm{Cl}}^{-}$ at pH = 2 in Figure 13a. The interaction energies for both ions with the membrane approach zero for *z* < −50 Å, because the ions are relatively far away from the membrane in the feed reservoir. As the applied body force biases the ions to move in the +*z* direction toward the membrane, the interaction energy becomes increasingly positive for ${\mathrm{Na}}^{+}$ and negative for ${\mathrm{Cl}}^{-}$. The positive ${\mathrm{Na}}^{+}$–membrane interaction energy indicates repulsive interactions with NH+2 groups in the membrane, and it becomes larger as ${\mathrm{Na}}^{+}$ ions move closer to the membrane surface. This repulsion peaks at the region with highest NH+2 density, indicated by the green curve below the interaction energy curves. Once the ${\mathrm{Na}}^{+}$ ions move past the densest region of NH+2, the interaction energy decreases, but it remains positive through the entire membrane.

Recalling that the interaction curves are overlaid for all $30\;{\mathrm{Na}}^{+}$ ions, it is evident that the interaction energy is quite similar in all cases. Fewer overlapping curves for *z*> 0 Å reflects the small number of ${\mathrm{Na}}^{+}$ ions that make it past the center of the membrane, consistent with the distribution of ${\mathrm{Na}}^{+}$ ions at 100 ns in Figure 7$A_{5}$. In fact, only one ${\mathrm{Na}}^{+}$ ion enters the permeate reservoir, reflected in the single curve past *z* = 20 Å. In contrast, ${\mathrm{Cl}}^{-}$ ions are attracted to NH+2, so the interaction energy becomes more negative as ${\mathrm{Cl}}^{-}$ moves closer to NH+2 charged groups. Just like ${\mathrm{Na}}^{+}$ ions, the interaction energy for ${\mathrm{Cl}}^{-}$ ions peaks where NH+2 groups are densest and then decreases as the ${\mathrm{Cl}}^{-}$ ions move through the membrane. Since several ${\mathrm{Cl}}^{-}$ ions make it to the permeate reservoir, there are more overlaid ${\mathrm{Cl}}^{-}$ ion curves to the right of *z* = 20 Å than for the single ${\mathrm{Na}}^{+}$ ion.

The interaction energy results for ${\mathrm{CaCl}}_{2}$ at pH = 2 are illustrated in Figure 13d. The ${\mathrm{Cl}}^{-}$–NH+2 interaction energy landscape does not differ from the NaCl case because the interaction energy only reflects the interaction energy between the selected solute ion and the membrane charges. However, the interaction strength depends on the valency of the cation such that ${\mathrm{Ca}}^{2+}$ exhibits roughly two times the interaction energy of ${\mathrm{Na}}^{+}$. Furthermore, the interaction energy curves do not extend as far into the membrane for ${\mathrm{Ca}}^{2+}$ and ${\mathrm{Cl}}^{-}$ in the ${\mathrm{CaCl}}_{2}$ case because the ions do not make it as far into the membrane.

Analogous results occur when the membrane charge contains only negatively charged ${\mathrm{COO}}^{-}$ groups at pH = 10, as shown in Figure 13c,f, although the sign of the interaction energy is reversed and the curves do not extend as far to the right due to absence of feed ions traversing traversing very far in the +*z* direction. That is, for a negatively charged membrane, the ${\mathrm{Cl}}^{-}$ ions have a positive interaction energy, reflecting the repulsive interaction with ${\mathrm{COO}}^{-}$ inside the membrane. On the other hand, ${\mathrm{Na}}^{+}$ ions have an attractive interaction with ${\mathrm{COO}}^{-}$ in the membrane, so the interaction energy is negative and peaks at the densest region of ${\mathrm{COO}}^{-}$. Note the maximum magnitude of the interaction energy at pH = 10 is about twice as large as that at pH = 2 because there are about twice as many ${\mathrm{COO}}^{-}$ groups at pH = 10 as NH+2 groups at pH = 2. In addition, the peak in the ${\mathrm{Na}}^{+}$–${\mathrm{COO}}^{-}$ energy is further to the right at pH = 10 because more ${\mathrm{COO}}^{-}$ sites are located toward the permeate reservoir side of the membrane. Similar results occur for ${\mathrm{Ca}}^{2+}$ and ${\mathrm{Cl}}^{-}$ ions, as shown in Figure 13f, except that the ${\mathrm{Ca}}^{2+}$ interaction energy is larger than that for ${\mathrm{Na}}^{+}$ due to the valency of ${\mathrm{Ca}}^{2+}$ ions. In this case, the ions do not penetrate very far into the membrane, so the interaction energy curves do not extend much past *z* = −10 Å.

When both NH+2 and ${\mathrm{COO}}^{-}$ groups are present at pH = 7, the interaction energy landscape reflects interactions of the ions depending on the locations of the charged groups in the membrane, as shown in Figure 13b,e. The feed ions first encounter a high concentration of ${\mathrm{COO}}^{-}$ groups near the membrane feed surface, reflected as a positive (repulsive) interaction energy for ${\mathrm{Cl}}^{-}$ ions and a negative (attractive) interaction energy for ${\mathrm{Na}}^{+}$ ions and about twice as large of a negative energy for ${\mathrm{Ca}}^{2+}$ ions. Neither of the cations make it far into the membrane, but ${\mathrm{Cl}}^{-}$ ions make it through the membrane into the reservoir. In fact, the attractive interaction of ${\mathrm{Cl}}^{-}$ ions with NH+2 ions seems to aid ${\mathrm{Cl}}^{-}$ ion transport, evident as the slightly negative (attractive) interaction energy for *z* > 20 Å. These results are consistent with previous measurements of the interaction energy for ${\mathrm{Cl}}^{-}$, ${\mathrm{Li}}^{+}$, and ${\mathrm{Mg}}^{2+}$ through NF membranes [4].

To further elucidate the impact of membrane charge group distribution on ion transport, we also examine the interaction energy landscape for the five alternative membrane systems at pH = 7, as shown in Figure 14. One might expect the interaction energy for the flip case shown in Figure 14a,f to mirror the pH = 7 case shown in Figure 13b,e, because it is the same membrane with the feed and permeate sides flipped. However, this is not the case because the ions encounter different membrane charge groups upon first entering the membrane in the two cases. In the flip case, ${\mathrm{Na}}^{+}$ ions pass through the repulsive NH+2 feed surface layer easily, drawn further into the membrane by the attractive ${\mathrm{COO}}^{-}$ groups on the permeate side of the membrane, while ${\mathrm{Cl}}^{-}$ ions are stuck at the feed surface due to attractive interactions with NH+2. Nevertheless, based on the symmetry of the charge distribution, we would expect that the interaction energy curves for the ions in the range of *z* = −50 to *z* = 0 Å in the flip case would complement the missing portion in the control case for *z* = 0 Å to *z* = 50 Å.

The bilayer case in Figure 14b is similar to the pH = 7 case in Figure 13b. That is, ${\mathrm{Na}}^{+}$ ions are stuck at the feed surface due to ${\mathrm{COO}}^{-}$ groups, while ${\mathrm{Cl}}^{-}$ ions pass through this repulsive region, drawn by the attractive interactions with NH+2 groups on the permeate side of the membrane. The situations for the flip and bilayer cases for divalent ${\mathrm{Ca}}^{2+}$ ions is less interesting because no ions get very far across the membrane during the 100 ns simulation.

The cases without NH+2 groups in the membrane models (Figure 14c–e,h–j) are less interesting. The interaction energy curves correlate directly with the ${\mathrm{COO}}^{-}$ distribution and the solute feed charges, as expected. Note that the maximum in the interaction energy, whether positive or negative, always aligns with the maximum in the membrane charge group distribution, whether NH+2 or ${\mathrm{COO}}^{-}$, in all cases shown in Figure 13 and Figure 14. Clearly, charge distribution affects the passage of ions through the membrane, although steric and other effects also play key roles.

This analysis is by no means a flawless measure of the interactions between solute ions and the membrane, because it reflects the interaction energy between two entities in a vacuum environment without accounting for the effects of water, solvation energy, and steric interactions. In fact, solvation can significantly dampen interactions between charged entities, often by an order of magnitude [81,82,83,84,85], which is not reflected in this analysis. Regardless, this measurement of the interaction energy allows a direct comparison between different types of feed ions that reflects their relative electrostatic interactions with the overall membrane charge and the specific details of the membrane charge location.

It is tempting to correlate the solute–membrane interaction energies with the macroscopic concept of charge-based selectivity through Donnan exclusion. Based on classical Donnan exclusion principles, the fixed charged groups in the membrane facilitate the partition of solutes bearing an opposite charge (counter-ions) into regions near these fixed charge groups while impeding the entry of similarly-charged ions (co-ions). This leads to a state of thermodynamic equilibrium between the ions in the feed solution and those within the active layer. According to Donnan partitioning theories, distinct energy barriers and wells are associated with the interactions between the membrane charge groups and both co-ions and counter-ions. However, the specific dynamics of these energy barriers and wells have not been clearly established, particularly at the single ion and single membrane charged group level. The results in this section and previous sections show clearly that ion–membrane charge interactions can have significant impact on the transport of ions through the membrane. However, while counter-ions are indeed impeded by interactions with charged groups in the membrane, co-ions are not very strongly affected. Nevertheless, further work is needed. Of particular value would be long duration simulations that reflect equilibrium conditions. However, such simulations are well beyond the scope of this study.

#### 3.3.2. Ion Hydration in Membranes

The hydration of ions due to the strong electrostatic interaction between an ion as a point charge and water dipoles is also thought to be important for ion rejection [7,38,64]. Water molecules form a hydration shell surrounding an ion, making it more difficult for the ion to pass through the intermolecular space in the membrane. To better understand the impact of the hydration shell on ion interactions in the membrane, we calculate the water coordination number, which is the average number of water molecules within the first solvation shell of the ions, as the ions permeate through the membrane. To determine which water molecules to include in the hydration shell, the radial distribution function (RDF) is calculated for all ion–water oxygen pairs [66], and the cutoff distance for water molecules in the first hydration shell is set as the radius in the RDF corresponding to the first valley after the first peak in the RDF. The cutoff distance is 3.0 Å for ${\mathrm{Na}}^{+}$, 3.0 Å for ${\mathrm{Ca}}^{2+}$, and 3.6 Å for ${\mathrm{Cl}}^{-}$, similar to other recent studies [49,86,87].

The dependence of the water coordination number on the position of ions in the membrane is shown in Figure 15 for pH = 2, 7, and 10. Each curve represents the average water coordination number for all ions over all three trials for all positions in the reservoirs and the membranes. In several cases, the curves do not extend through the entire membrane because no ions permeate further than the right end of the curve. In all cases, the coordination number decreases as the ion moves from the feed solution into the membrane, as would be expected based on the RDF cutoff distances of 3.0 Å and 3.6 Å being larger than the mean pore radius of 2.8 Å. For all three pH levels, the coordination number for ${\mathrm{Ca}}^{2+}$ in the feed reservoir is higher than for ${\mathrm{Na}}^{+}$ because of its greater charge. However, on the feed side of the membrane, the ${\mathrm{Ca}}^{2+}$ coordination number drops off more quickly than that for ${\mathrm{Na}}^{+}$, and it is at a slightly lower level within the membrane. The large, sharp drop-off in the coordination number for ${\mathrm{Ca}}^{2+}$ is accentuated for pH = 7, most likely due to the high density of ${\mathrm{COO}}^{-}$ groups near the feed surface (note the distributions for charged groups in the lower part of each figure).

Two curves for ${\mathrm{Cl}}^{-}$ at each pH reflect results for NaCl and ${\mathrm{CaCl}}_{2}$. The ${\mathrm{Cl}}^{-}$ coordination number for ${\mathrm{CaCl}}_{2}$ drops off more quickly than that for NaCl at the feed surface, similar to the trends for the respective cations. This suggests that cation–anion interactions at the feed surface result in the shedding of hydrated water molecules. This occurs regardless of pH, indicating that it is unrelated to membrane charge.

The coordination number curves are similar for each ion regardless of pH (see overlaid coordination number curves for all pH values in Appendix A). The only exception is the slightly slower drop-off for ${\mathrm{Ca}}^{2+}$ ions at pH = 10, mostly likely because of the buildup of ${\mathrm{Ca}}^{2+}$ ions at the feed surface of the membrane. Nevertheless, the similarity in the coordination number across the various charge distributions associated with each pH level suggests that membrane charge plays a small role in the number of water molecules associated with ions as they permeate through the membrane.

#### 3.3.3. Ion Diffusivity in Membranes

Another metric to quantify solute ion permeance at the molecular scale is the ion diffusivity based on the mean square displacement (MSD) of the solute ions when they are inside the membrane, similar to our previous approach for water diffusivity within the membrane [49]. Diffusivity is related to the MSD using Einstein’s theory of diffusion [88],(2)$$MSD\left(\tau \right)=2nD_{\alpha }\tau^{\alpha }$$
where τ is the time interval for the MSD calculation, *n* is the number of dimensions for the motion of the ions (*n*
=3 in this study), α represents the anomalous diffusion exponent characterizing the degree to which the diffusivity deviates from normal diffusion [89,90], and $D_{\alpha }$ is the generalized diffusion coefficient. What are important here are both $D_{\alpha }$ as a relative measure of the diffusivity and α as a relative measure of how much ions are hindered in their motion by electrostatic and steric effects compared to normal diffusive behavior, where α=1.

MSD curves for NaCl and ${\mathrm{CaCl}}_{2}$ are shown in Figure 16 for the membrane models in Figure 3a–c, corresponding to pH = 2, 7, and 10. The solid line is the average MSD over the three simulations and the shaded region represents the range of MSD values at a given value of τ. The higher MSD curve for ${\mathrm{Cl}}^{-}$ ions indicates that the diffusion of ${\mathrm{Cl}}^{-}$ ions is slightly greater than that of either cation in all cases, consistent with previous measurements of the diffusion coefficient in bulk water [66,91] and in the concentration polarization layer of a NF270 membrane [92]. However, the diffusion of ${\mathrm{Cl}}^{-}$ is always related to the diffusion of the cation. For example, as the pH increases, the ${\mathrm{Cl}}^{-}$ MSD curve is always just above the ${\mathrm{Na}}^{+}$ curve even though the ${\mathrm{Na}}^{+}$ curve drops lower as the pH increases.

Since the MSD curves are not linear, we select a specific range of τ to measure α [93]. To determine α, we fit a line to the MSD for the range 0.03≤τ≤0.1 ns, which corresponds to that used previously for measuring water diffusivity in similar membrane simulations [49]. Values for α, which corresponds to the slope of the curve over this range of τ, are indicated next to the MSD curves. The slopes of MSD–τ curves are generally less than 1, indicating a subdiffusive [89] situation for both cations and anions. This is expected, because the ion motion is impeded by membrane charge and by steric effects. These constraints are most evident for ${\mathrm{CaCl}}_{2}$, as would be expected since both ions are unable to transit very far past the feed side of the membrane. The values of α are larger for the NaCl system. The only case that is not subdiffusive is that for ${\mathrm{Na}}^{+}$ ions at pH =7, although we attribute this to the particular range of τ considered and the relative difficulty in measuring α, rather than to any physical effect.

A specific value for τ must be selected to estimate the relative diffusivity, *D_α_*, which we express here in terms of the MSD value (based on the relation between the MSD and *D_α_* in Equation (Equation 2)). Here, we use τ=0.06 ns, which is at the approximate midpoint of the range used to measure α to be certain to avoid smaller τ where pore size effects play a role and larger τ where the simulated system size plays a role [49]. We indicate the MSD value, which is proportional to *D_α_*, via horizontal dotted lines in Figure 16 to allow comparison of the relative magnitude of the diffusion between the various pH conditions.

The diffusion of the individual ions for ${\mathrm{CaCl}}_{2}$ is always less than that for NaCl at the same pH, and for both NaCl and ${\mathrm{CaCl}}_{2}$, the diffusion at pH = 10 is less than the diffusion at pH = 2 and 7. Furthermore, the difference in the diffusion between the cation and the anion is smaller in ${\mathrm{CaCl}}_{2}$ solutions compared to NaCl solutions. This further indicates that ${\mathrm{Ca}}^{2+}$ ions often associate or cluster with ${\mathrm{Cl}}^{-}$ ions in the membrane. It is hard to draw any specific conclusions beyond this, and we are not aware of any experimental results directly measuring the diffusivity of ions inside the membrane at the pH values used in this study. Also note that the MSD curves have varying widths of the shaded regions, which is due to variations in the numbers of ions available in the membrane for sampling at each τ. Nevertheless, the MSD results are consistent with the results in Figure 7 and Figure 8.

## 4. Conclusions

In this study, we have advanced the understanding of solute transport in charged polyamide membranes by incorporating realistic membrane charge characteristics, both ${\mathrm{COO}}^{-}$ and NH+2 functional groups, depending on pH. By employing body force-biased molecular dynamics (MD) simulations, we can compare the transport behaviors of monovalent and divalent cations. The results reveal distinct transport mechanisms: Divalent cations tend to associate with their pairing anions and are hindered by size exclusion, leading to their accumulation on the membrane surface. In contrast, monovalent cations exhibit weaker interactions with their pairing anions, allowing for increased ion passage. While the energetics analysis captures electrostatic interactions and the time series plots of ion location allude to steric interactions, there is also potential for chelation between ${\mathrm{Ca}}^{2+}$ and ${\mathrm{COO}}^{-}$ groups, which could further limit ${\mathrm{Ca}}^{2+}$ mobility compared to monovalent ions [44,45]. Future studies using quantum mechanics methods may help clarify the role of specific ion–ligand coordination in these systems.

Our findings emphasize the critical roles of membrane charge location and local density in ion transport, especially under conditions where steric repulsion influences solute behavior, including the clustering of ions with their counter-ions. Depending on the pH, ions may be either rejected by steric repulsion or attracted by the membrane charge. Specifically, the presence of a ${\mathrm{COO}}^{-}$ barrier on the membrane surface impedes divalent cation transport. The energetics measurements between solute ions and the membrane further amplify the importance of the underlying impact of electrostatics attraction and repulsion. Notably, membrane charge seems to have a more profound impact on counter-ion transport than on co-ion transport. Counter-ions tend to associate with membrane charged groups due to electrostatic attraction, while co-ions, which are repulsed by the membrane charged groups, seem to pass by the membrane charged groups having the same charge, especially if there are attractive membrane charges deeper in the membrane.

This study sets the stage for the continuing study of solute–membrane charge interactions. These and future insights will help inform the design of next-generation NF and RO membranes that are more efficient and selective. The research described here extends our previous work where we considered transport of ${\mathrm{Li}}^{+}$ and ${\mathrm{Mg}}^{2+}$ ions through NF membranes [4]. While we focus on polyamide NF membranes here, many of the insights from this research carry over to ion–membrane charge interactions for RO membranes and other charged membranes, as well. This will likely have implications for the design and optimization of advanced NF and RO membranes by providing a more comprehensive understanding of how specific ions interact with different functional groups within the membrane, perhaps extending to tailored ion-rejection properties. This aspect of membrane design is particularly relevant in light of the work by Elvati and Violi, who demonstrated the utility of MD simulations in understanding permeant–membrane interactions [94]. Moreover, our findings contribute to the ongoing discourse on the role of nano-inclusions in membrane science, as explored by Lemaalem et al. [95].

Although this study enhances the understanding of ion–membrane interactions from a molecular-level perspective, it represents only a portion of the overall picture. Much work remains. For instance, longer simulation times are needed to observe charge equilibrium in the permeate reservoir. As such, non-equilibrium simulations with a transmembrane pressure and a thicker membrane would make the simulations more realistic. Of course, even longer simulations may make it possible to avoid the need to apply an external force to artificially drive the ions through the membrane. In addition, we model the charged groups as fixed for the duration of the simulation, thereby neglecting the random ongoing protonation and deprotonation of the amine and carboxyl groups. The impact of this simplifying assumption remains unclear. Furthermore, the results are also dependent on the MD models used for water and other interactions [66,96]. Moreover, as is evident in Section 2.1, a better understanding of the actual distribution of negative and positive charges in real membranes would improve simulation accuracy. In spite of these limitations, this research provides a novel framework for better understanding ion–membrane charge interactions and evaluating established macroscopic theories like Donnan exclusion. Beyond membrane applications in water treatment, the insights and simulation methods in this study may also be useful for other membrane applications as well as for systems such as solid-state nanopores, biosensors, and electrochemical devices where ion–surface charge interactions are important. Further studies using MD simulations have the potential to significantly impact the field of membrane technology, paving the way for more efficient and targeted membrane designs.

## Figures and Tables

**Figure 1 membranes-15-00184-f001:**
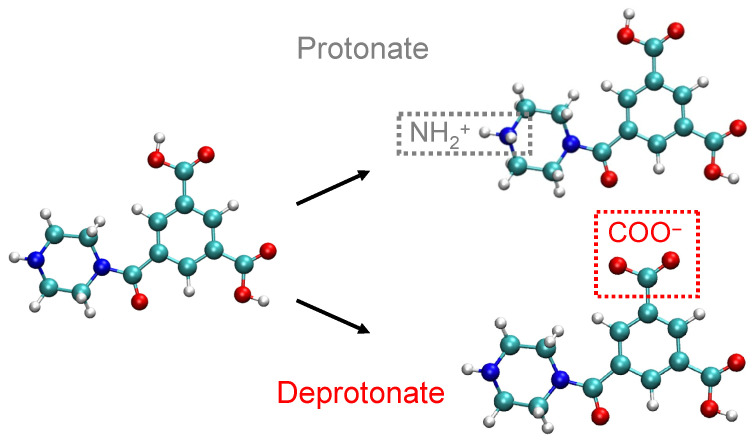
A polymerized PIP-TMC dimer undergoes protonation (**top**) forming an ionized amine group (*R*-NH+2) and deprotonation (**bottom**) forming a carboxylate group (*R*-${\mathrm{COO}}^{-}$). Blue, white, cyan, and red beads represent N, H, C, and O, respectively. In some cases, H atoms are behind N or C atoms, and in other cases, H atoms overlay C or N atoms (for example, see NH+2).

**Figure 2 membranes-15-00184-f002:**
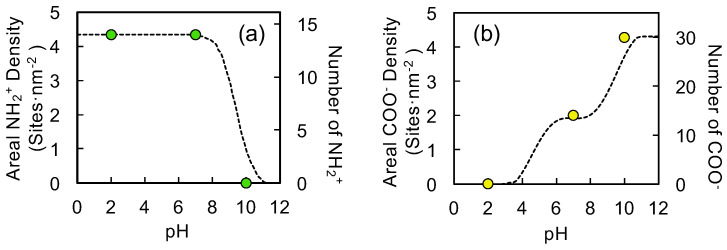
Dependence on pH for the areal density (left vertical axis) of (**a**) ionized amine groups and (**b**) carboxylate end groups and the number of corresponding charged end groups (right vertical axis) in our membrane model of an NF270 membrane [18]. Data points represent experimental values from Ritt et al. [18] and curves represent fits to experiment data provided by Ritt et al.

**Figure 3 membranes-15-00184-f003:**
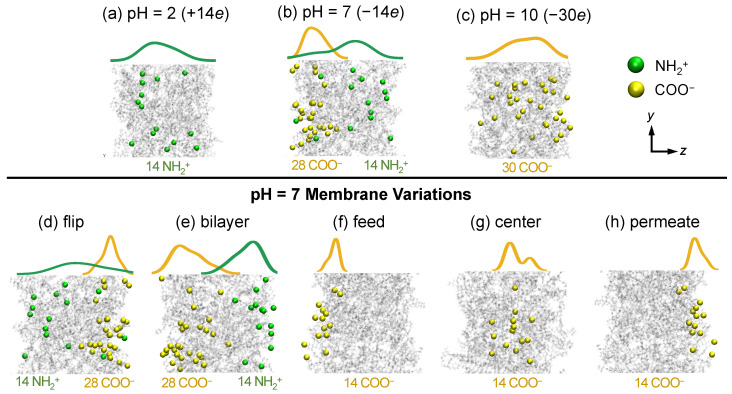
Membrane models at various pH conditions reflecting the membrane’s net charge. (**a**) pH = 2 (+14*e*), (**b**) pH = 7 (−14*e*), and (**c**) pH = 10 (−30*e*), and (**d**–**h**) pH = 7 with membrane net charge of −14*e*. Carboxylate and ionized amine groups are represented as yellow and green beads in the membrane model representations, respectively. Charged groups are shown in the *yz*-plane but are actually distributed in the *x*-direction as well. The distribution of charged groups through the thickness of the membrane is indicated qualitatively just above each membrane model (yellow curves for ${\mathrm{COO}}^{-}$ and green curves for NH+2).

**Figure 4 membranes-15-00184-f004:**
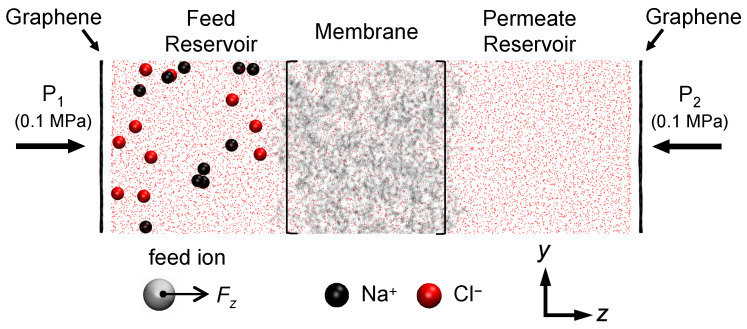
Body force-induced solute transport simulation setup. Water molecules are red points, graphene sheets are black, the membrane is represented as a light gray matrix, with the approximate location of the feed and permeate surfaces indicated by black brackets, feed sodium ions are black beads, and feed chlorine ions are red beads. Water molecule points and ion beads are not to scale.

**Figure 5 membranes-15-00184-f005:**
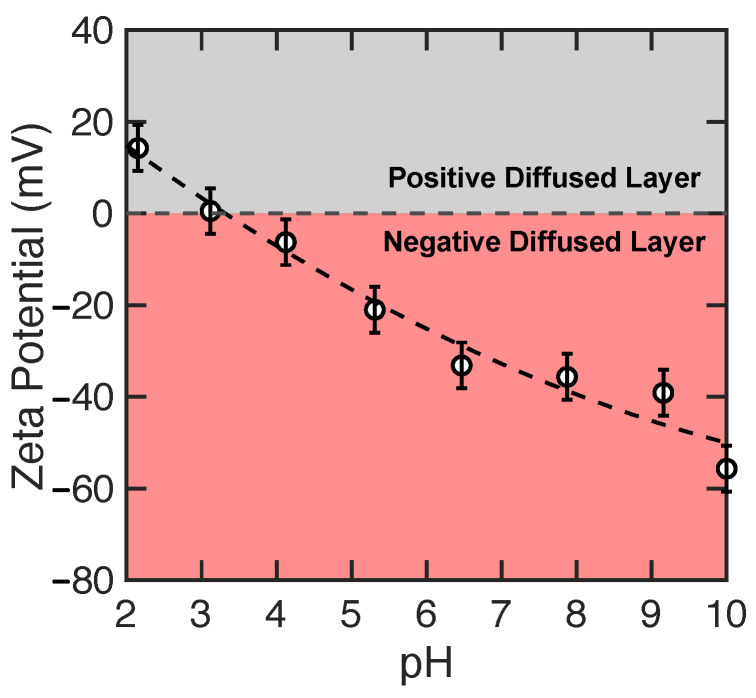
Zeta potential as a function of solution pH for NF270. A 0.1 M KCl solution is used as the electrolyte stream, and the zeta potentials (circles) are calculated from the streaming potential with the Smoluchowski equation [67,68]. The dashed curve is based on a second-order polynomial interpolation. The error bars are standard deviations from triplicate measurements.

**Figure 6 membranes-15-00184-f006:**
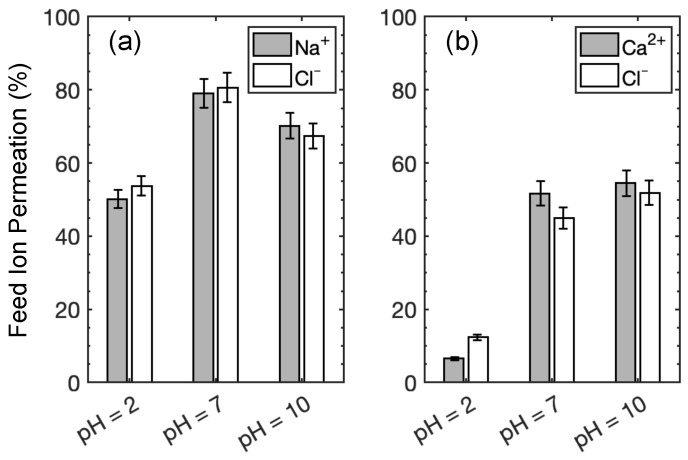
Feed ion permeation *$P_{i}$* as a function of solution pH for (**a**) NaCl and (**b**) ${\mathrm{CaCl}}_{2}$. The error bars are standard deviations from triplicate measurements.

**Figure 7 membranes-15-00184-f007:**
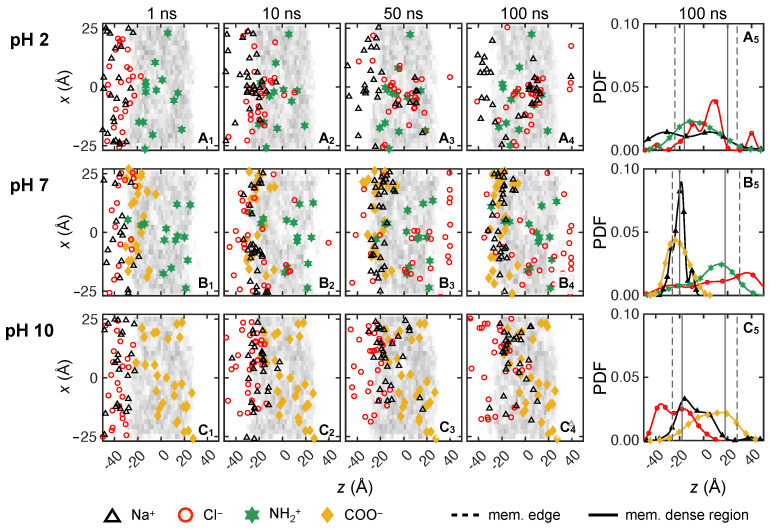
Feed ion locations of 0.133 M NaCl solutions in the *xz*-plane for left-to-right transport across three trials for (**A**) pH = 2, (**B**) pH = 7, and (**C**) pH = 10 at (**1**) 1 ns, (**2**) 10 ns, (**3**) 50 ns, and (**4**) 100 ns. Ion and charged group locations are projected onto the *xz*-plane, although they are at varying positions in the depthwise *y*-direction. Gray areas indicate the local membrane density, where the darker gray implies higher density. (**5**) Probability density function of feed ions and membrane functional groups (${\mathrm{COO}}^{-}$, NH+2) in the *z* direction at 100 ns. Symbols on the distribution curves are merely to identify the curve; they do not represent individual data points. Dashed lines mark the edge-to-edge boundaries of the membrane model, and solid lines bound the densest regions of the membrane. For all pH levels depicted here, NH+2 and ${\mathrm{COO}}^{-}$ are represented by solid green stars and solid yellow diamonds, respectively. ${\mathrm{Na}}^{+}$ and ${\mathrm{Cl}}^{-}$ are indicated by hollow black triangles and red circles, respectively. The *x*-axis is stretched compared to the *z*-axis to more clearly show the distribution of ions and charged end groups.

**Figure 8 membranes-15-00184-f008:**
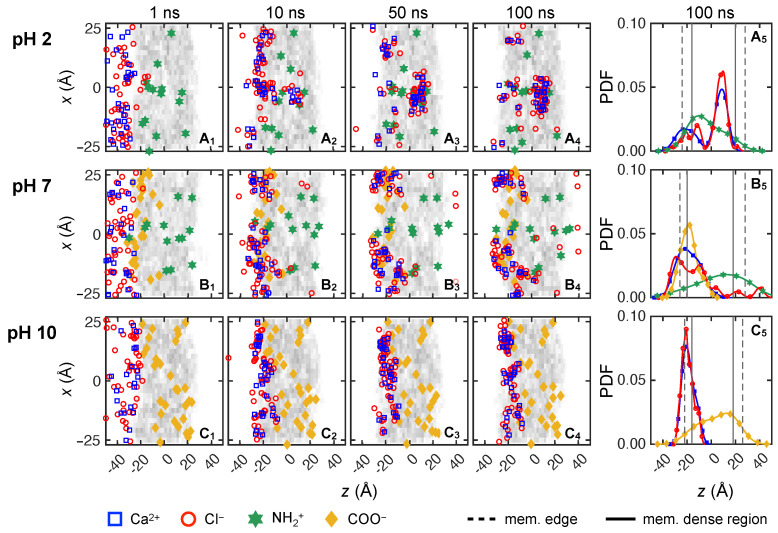
Feed ion locations of 0.133 M ${\mathrm{CaCl}}_{2}$ solutions in the *xz*-plane for left-to-right transport across three trials for (**A**) pH = 2, (**B**) pH = 7, and (**C**) pH = 10 at (**1**) 1 ns, (**2**) 10 ns, (**3**) 50 ns, and (**4**) 100 ns. Ion and charged group locations are projected onto the *xz*-plane, although they are at varying positions in the depthwise *y*-direction. Gray areas indicate the local membrane density, where the darker gray implies higher density. (**5**) Probability density function of feed ions and membrane functional groups (${\mathrm{COO}}^{-}$, NH+2) in the *z* direction at 100 ns. Symbols on the distribution curves are merely to identify the curve; they do not represent individual data points. Dashed lines mark the edge-to-edge boundaries of the membrane model, and solid lines bound the densest regions of the membrane. For all pH levels depicted here, NH+2 and ${\mathrm{COO}}^{-}$ are represented by solid green stars and solid yellow diamonds, respectively. ${\mathrm{Ca}}^{2+}$ and ${\mathrm{Cl}}^{-}$ are indicated by hollow blue squares and red circles, respectively. The *x*-axis is stretched compared to the *z*-axis to more clearly show the distribution of ions and charged end groups.

**Figure 9 membranes-15-00184-f009:**
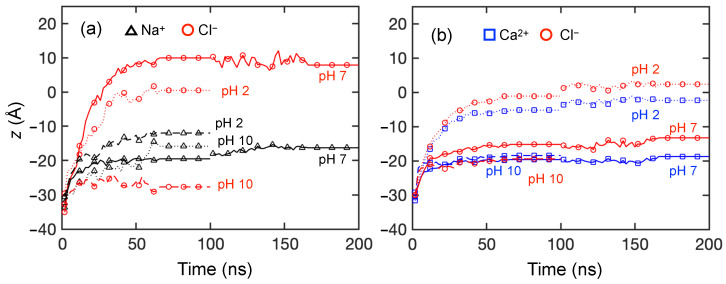
Ensemble-averaged feed ion *z*-locations throughout the 100 ns simulations across 3 trials within membrane systems at pH = 2, pH = 7, and pH = 10 using (**a**) NaCl solute feed and (**b**) ${\mathrm{CaCl}}_{2}$ solute feed. Additionally, NaCl at pH 7 and ${\mathrm{CaCl}}_{2}$ at pH 2 and pH 7 are extended to 200 ns. ${\mathrm{Na}}^{+}$, ${\mathrm{Ca}}^{2+}$, and ${\mathrm{Cl}}^{-}$ are indicated by hollow black triangles, blue squares, and red circles, respectively.

**Figure 10 membranes-15-00184-f010:**
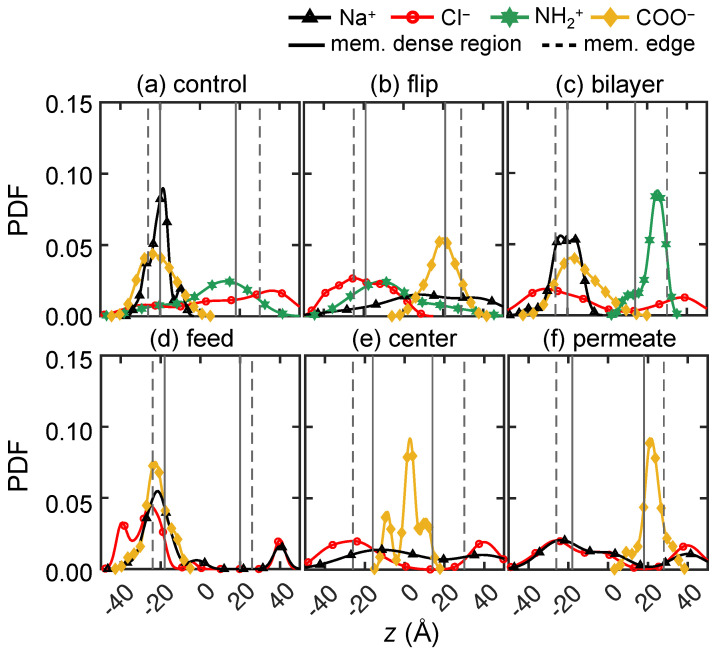
The probability density function of feed ions and membrane functional groups (${\mathrm{COO}}^{-}$ and NH+2) in the *z* direction at *t* = 100 ns using 6 variations of membrane models, each with a net charge of −14*e* but with different charge distributions defined in Figure 3: (**a**) control, (**b**) flip, (**c**) bilayer, (**d**) feed, (**e**) center, and (**f**) permeate. Symbols on the distribution curves are merely to identify the curve; they do not represent individual data points. Dashed lines mark the edge-to-edge boundaries of the membrane model, and solid lines bound the densest regions of the membrane. ${\mathrm{Na}}^{+}$, ${\mathrm{Cl}}^{-}$, NH+2, and ${\mathrm{COO}}^{-}$ are indicated by black triangles, red circles, green stars, and yellow diamonds, respectively.

**Figure 11 membranes-15-00184-f011:**
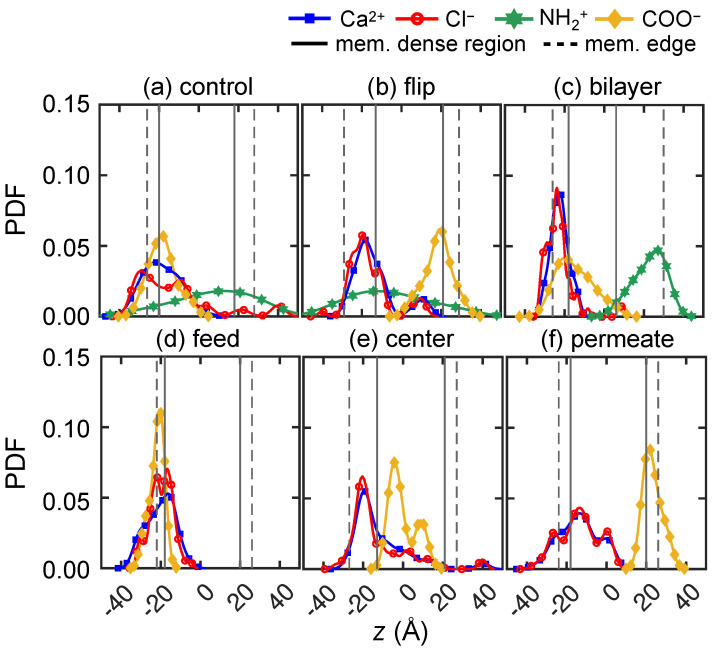
The probability density function of feed ions and membrane functional groups (${\mathrm{COO}}^{-}$ and NH+2) in the *z* direction at *t* = 100 ns using 6 variations of membrane models, each with a net charge of −14*e* but with different charge distributions defined in Figure 3: (**a**) control, (**b**) flip, (**c**) bilayer, (**d**) feed, (**e**) center, and (**f**) permeate. Symbols on the distribution curves are merely to identify the curve; they do not represent individual data points. Dashed lines mark the edge-to-edge boundaries of the membrane model, and solid lines bound the densest regions of the membrane. ${\mathrm{Ca}}^{2+}$, ${\mathrm{Cl}}^{-}$, NH+2, and ${\mathrm{COO}}^{-}$ are indicated by blue squares, red circles, green stars, and yellow diamonds, respectively.

**Figure 12 membranes-15-00184-f012:**
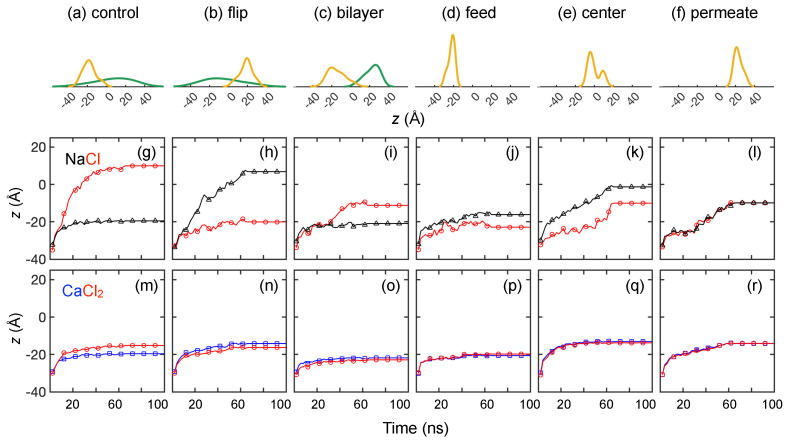
Ensemble-averaged feed ion *z*-locations throughout the 100 ns simulations across 3 trials within membrane systems with a net charge of ${\mathrm{14e}}^{-}$ but with different charge distributions as defined in Figure 3. Probability density function of ${\mathrm{COO}}^{-}$ (yellow) and NH+2 (green) in the z direction for (**a**) control, (**b**) flip, (**c**) bilayer, (**d**) feed, (**e**) center, and (**f**) permeate. (**g**–**l**) NaCl solute feed; (**m**–**r**) ${\mathrm{CaCl}}_{2}$ solute feed. ${\mathrm{Na}}^{+}$, ${\mathrm{Ca}}^{2+}$, and ${\mathrm{Cl}}^{-}$ are indicated by hollow black triangles, blue squares, and red circles, respectively.

**Figure 13 membranes-15-00184-f013:**
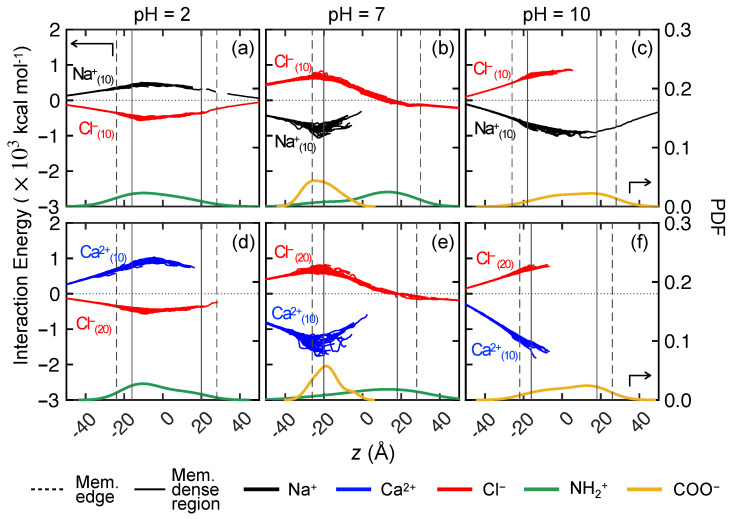
Interaction energy between feed ions and the membrane in the *z*-direction of membrane thickness across all trials at different pH values for ${\mathrm{Na}}^{+}$ (black) and ${\mathrm{Cl}}^{-}$ (red) using (**a**–**c**) NaCl feed solutions and ${\mathrm{Ca}}^{2+}$ (blue) and ${\mathrm{Cl}}^{-}$ (red) using (**d**–**f**) ${\mathrm{CaCl}}_{2}$ feed solutions. Probability density functions of NH+2 (green) and ${\mathrm{COO}}^{-}$ (yellow) groups are shown in the lower portion of each panel. Dashed vertical lines indicate the edge-to-edge boundaries of the membrane model, and solid vertical lines mark the densest regions of the membrane. Numbers in parentheses indicate the number of ions in each of 3 trials.

**Figure 14 membranes-15-00184-f014:**
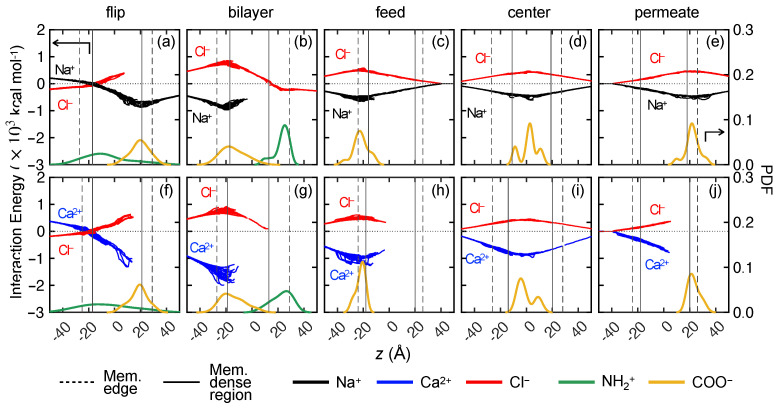
Interaction energy between feed ions and the membrane in the *z*-direction of membrane thickness across all trials in 5 membrane variations for ${\mathrm{Na}}^{+}$ (black) and ${\mathrm{Cl}}^{-}$ (red) using (**a**–**e**) NaCl feed solutions and ${\mathrm{Ca}}^{2+}$ (blue) and ${\mathrm{Cl}}^{-}$ (red) using (**f**–**j**) ${\mathrm{CaCl}}_{2}$ feed solutions. Probability density functions of NH+2 (green) and ${\mathrm{COO}}^{-}$ (yellow) groups are shown in the lower portion of each panel. Dashed vertical lines indicate the edge-to-edge boundaries of the membrane model, and solid vertical lines mark the densest regions of the membrane.

**Figure 15 membranes-15-00184-f015:**
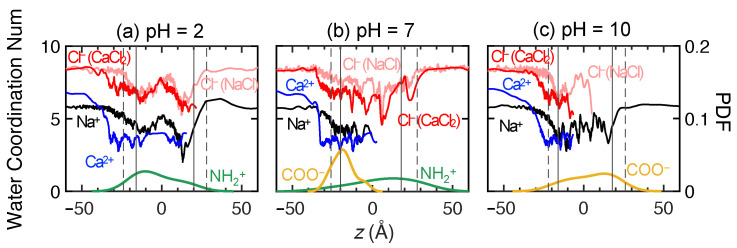
Water coordination number in the *z*-direction of membrane thickness for left-to-right transport across all trials for NaCl and ${\mathrm{CaCl}}_{2}$ feeds at (**a**) pH = 2, (**b**) pH = 7, and (**c**) pH = 10, where black curves indicate ${\mathrm{Na}}^{+}$ ions, blue for ${\mathrm{Ca}}^{2+}$, pink for ${\mathrm{Cl}}^{-}$ in NaCl, and red for for ${\mathrm{Cl}}^{-}$ in ${\mathrm{CaCl}}_{2}$. Probability density functions of NH+2 (green) and ${\mathrm{COO}}^{-}$ (yellow) groups are shown in the lower portion of each panel. Dashed vertical lines indicate the edge-to-edge boundaries of the membrane model, and solid vertical lines mark the densest regions of the membrane.

**Figure 16 membranes-15-00184-f016:**
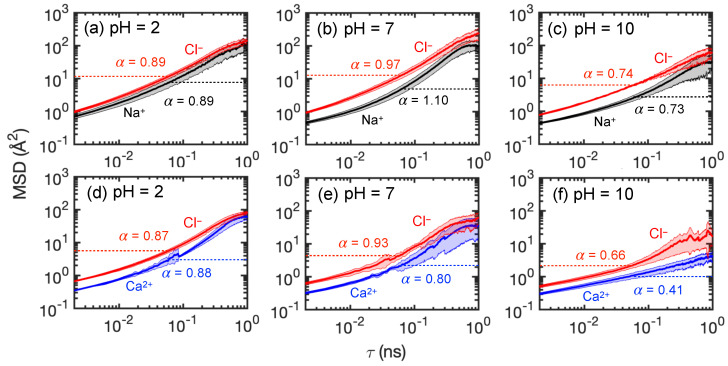
Meansquare displacement (MSD) of feed ions within a membrane as a function of the time interval (τ) in a NaCl feed at (**a**) pH = 2, (**b**) pH = 7, and (**c**) pH = 10, where black curves indicate MSD curves for ${\mathrm{Na}}^{+}$ and red for ${\mathrm{Cl}}^{-}$. Similarly, MSD as a function of τ measured for feed solute ions in a ${\mathrm{CaCl}}_{2}$ feed (**d**) pH = 2, (**e**) pH = 7, and (**f**) pH = 10, where blue curves indicate MSD curves for ${\mathrm{Ca}}^{2+}$ and red for ${\mathrm{Cl}}^{-}$. The shaded region represents the range of MSD values measured at a given τ. The dashed lines mark the MSD values for their associated ions at τ = 0.06 ns. The slope, α, for ensemble-averaged MSD curves for the range 0.03≤τ≤0.1 ns are also indicated.

## Data Availability

Input files for simulations and processed results from experiments will be available for at least 5 years after publication upon written request to the authors.

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
