# Peer review of "Membrane Charge Effects on Solute Transport in Nanofiltration: Experiments and Molecular Dynamics Simulations"

_membranes, 2025, doi:10.3390/membranes15060184_

Round 1
Reviewer 1 Report
Comments and Suggestions for Authors
In this paper, transport of CaCl2 and NaCl through nanofiltration membranes of differing charge distributions—based on changing pH conditions—are investigated primarily through molecular dynamics simulations. Experiments are used to confirm general trends. The authors thoroughly consider several charge distributions, which is of interest in designing novel nanofiltration membranes. They also probe different variables to explain their observations seen in the simulations, including interaction energy and water coordination numbers. The work provides interesting insights that can be expanded upon to inform future membrane formation for especially ion-ion selective membranes, which are important for critical resource recovery.
While this reviewer believes the work is valuable and should be published, some revisions to improve clarity and explain methodological rationale are warranted.
- Title: Giving an indication that this is based on molecular dynamics simulations on nanofiltration membranes would be helpful to communicate to the reader more specifically about the work. “Elucidating Membrane Charged Effects on Solute Transport in Nanofiltration Polyamide Membranes through Molecular Dynamics Simulations”
- Abstract: For experimentalists, the abstract is slightly misleading. “when the membrane has a higher concentration of negatively charged groups, corresponding to a higher pH” is confusing before reading the paper. It sounds as though either the pH is causing the formation of groups that can be negatively charged (actually synthesizing them) or the authors are describing the pH of the membrane itself. Rather, it is better explain in the summary as something like “when more carboxyl groups within the membrane are dissociated and negatively charged, which is expected when using feeds of higher pH.”
- Positioning work’s broader impact: The authors have anticipated some criticisms regarding the relatively shorter modeling time frame compared to real-life membrane operation. In the conclusion, they anticipate the value of this work will be that it can be built upon for future development of membranes. I was left wondering “So what?” at the end of the Introduction. The claim, “we consider both simulations and experiments in this paper to advance overall understanding” still leaves me wondering how this is practically helpful. A hint of what is stated in the conclusion would help motivate readers to read on, that this work is prerequisite for understanding membrane transport at longer time scales.
- Explanation for charge distribution: I commend the authors for their explanation of deciding how to distribute charges, based on prior work of Ritt et. al. That being said, I would like to know if there is any consideration by the authors as to how accurate zeta potential reflects the charge of the membrane? Other papers have shown that especially porous media and soft particles result in inaccurate charge predictions by zeta potential. (See https://doi.org/10.1021/la047049t for instance). The authors should address this. Regardless, any number of configurations, whether completely accurately representing the membrane’s pH response or not, are useful. How did the authors “randomly” distribute charges? Did they personally randomly select or did a computer?
- Accuracy of simulation (some possibly sources of inaccuracy that need addressing): The authors explain that the difference in experiment and simulation is likely the timescale and difference in membrane thicknesses. They also point out that charges are fixed while in reality, functional groups would be protonating and deprotonating. Although this is true, several other major differences are not addressed that should be: a.) The experimental work used pressure gradient as the driving force rather than a body force. What difference could this make? b.) The membrane formation does not result in a typical structure; polyamide membranes have large void spaces caused by temperature gradients during formation. Studies have shown that pore sizes are bimodal, including network and aggregate pores. How would this heterogeneity change things? c.) The mimicry of the pH conditions was achieved by changing charge distributions, but in reality, inducing pH variability changes the presence of OH-, H+, and any other ions within the agents used to change the pH; why was this not simulated in the feed conditions?
Comments on the Quality of English Language
A few grammar and clarity suggestions:
- Line 63: “membrane-based water treatment processes” is a misnomer for this work; should be “ion-ion aqueous separations in water”
- Line 316: cation concentration of 0.1 M mismatches the simulation conditions of 0.133 M; but it appears from Table S2 they actually matched?
- Line 472: states that the results of charge interactions are summarized in just one panel, but surely the writers meant the results for pH 7?
- Lines 489 – 490: somewhat redundant; “a hopping mechanism where Na+ ions can hop”
- Line 511: “only” repeated twice
- Line 686: “other” repeated twice
- Table S1: “flip” is repeated twice when I believe the authors meant “permeate” on the bottom configuration
Reviewer 2 Report
Comments and Suggestions for Authors
The manuscript by Liu et al. presents an innovative study on the molecular-level mechanisms of solute transport and rejection in polyamide nanofiltration (NF) membranes using molecular dynamics simulations. The authors examine the influence of membrane surface charge (COO⁻ and NH₂⁺) at varying pH levels on the transport of Na⁺ and Ca2+. The study finds that negatively charged membranes (higher pH) enhance Na⁺ and Cl⁻ rejection, while Ca²⁺ rejection remains consistently high for all the pH conditions. The simulations also reveal the importance of functional group distribution and steric effects in influencing ion transport. The findings are also validated by experimental data and offer valuable insights for the rational design of charged membranes with improved selectivity. Their results on how counter- and co-ions interact with charged surface groups are exciting. However, to be accepted for publication, below-sorted major and minor points need to be addressed by the authors.
Major Comments
In general, the manuscript is long and reading is tiring. For instance, in the final part of the intro, the authors introduce their work in a very detailed way.
The findings in this paper are not limited to only NF and RO membrane activities. The interactions between ions and charged surface groups are also important for other membranes, such as solid-state nanoporous membranes. To emphasize the importance of their findings, the authors should improve their introduction text by including further literature on electrochemical and optical methods that are used to probe ion-analyte interactions even from a single molecule level (DOI: 10.1039/D4MA00705K; DOI: 10.1039/D0CS01568G; https://doi.org/10.1021/nn303669g; https://doi.org/10.1002/admi.202201902).
In p3 – lines 118&119, authors claim that amine groups protonate, which depends on how far the pH is set from the IEP point! Does this mean that larger acidity will increase the number of deprotonated sites even beyond the buffer region of the IEP (pKa)? Is this because some surface (structural) deformations occur in the polymer at larger acidity? Then their model gets more complicated, I suppose? I suppose the authors believe the same happens for deprotonation too (the larger the pH, the more COO- sites). Then, how can the authors explain in Figure 6 that the Ca2+ permeation cannot go any higher than the permeation at pH7? This is maybe because the deprotonated sites cannot get any higher from pH7 to pH10.
One major challenge of the paper is the number of Cl- ions in NaCl (10 Cl- ions) and CaCl2 (20 Cl- ions) experiments. Here are my comments regarding to this issue:
p17 – lines 575-578, the Na+ ions are less associated with Cl- ions, and this might be because there are fewer Cl- ions in the media compared to the experiment with Ca2+. In Ca2+ experiments, 20 Cl- ions are associated with 10 Ca2+ ions. Can the authors comment on this?
Similarly, based on the arguments in p18 – lines 631 – 633, how can the Cl- - NH2+ interactions be similar in the two systems with different Cl- number (density)?
In p26 – lines 849-853, did the authors consider that maybe a larger number of Cl- ion in CaCl2 system makes the Cl- density towards the feed higher and thus the drop appears earlier?
In p28 – lines 906-909, the authors conclude their results as “monovalent cations exhibit weaker interactions with their pairing anions, allowing for increased ion passage.” can it also be that maybe Ca2+ ions having stronger interaction with COO- via chelation interactions that goes beyond a simple Coulombic attraction between the COO- and Ca2+. (see: https://advanced.onlinelibrary.wiley.com/doi/10.1002/admi.202201902; https://doi.org/10.1021/nn303669g). If so, this might also explain the more limited diffusion of Ca2+ than monovalent ions. The authors should comment on the possibility of such chelation interaction to extend their discussion beyond ionic interactions and steric repulsion reasons.
Minor comments/questions/suggestions:
- Methods: Add the names of the monomers (Piperazine (PIP) and trimesoyl chloride (TMC)) where they were mentioned for the first time in text (p3 – lines 98&99).
- The authors must explain the NF270 active layer in p4 – line 126.
- In Figure 4, the left graphene arrow should point the barrier rather than the feed solution.
- What are the concentrations of the HCl and NaOH (p9 – line316)?
- How did the authors generate the error bars in Figure 5 and Figure 6 ? This must be added to both the methods and the figure captions.
- In Figure 7, (especially at pH 7), the NH2+ (green) and COO- (orange) groups change slightly in their positions with increasing times (ns). How can this be possible?
- The reviewer could not understand how the ion pair lifetimes presented in p17 – 579 – 581 were generated.
- In Figure 16d, the shaded region of blue data looks strange (range of MSD goes suddenly to minimum and then increases again with larger time (tau)). Why is that?

Round 2
Reviewer 2 Report
Comments and Suggestions for Authors
The authors have thoroughly addressed all the comments raised in my initial review, resulting in a significantly improved manuscript that is now suitable for publication in Membranes.